# DecAlign: Hierarchical Cross-Modal Alignment for Decoupled Multimodal Representation Learning

**Chengxuan Qian[1], Shuo Xing[1], Shawn Li[2], Yue Zhao[2], Zhengzhong Tu[1]***

[1] Texas A&M University    [2] University of Southern California

## Abstract

Multimodal representation learning aims to capture both shared and complementary semantic information across multiple modalities. This intrinsic heterogeneity of diverse modalities presents substantial challenges to achieving effective cross-modal collaboration and integration. To address this, we introduce DecAlign, a novel hierarchical cross-modal alignment framework designed to decouple multimodal representations into modality-unique (heterogeneous) and modality-common (homogeneous) features. Specifically, we mitigate distributional discrepancies for modality-unique features via a novel prototype-guided optimal transport alignment strategy leveraging Gaussian mixture modeling and multi-marginal transport. Concurrently, semantic consistency across modalities is reinforced by aligning latent distribution matching with maximum mean discrepancy regularization. Furthermore, we incorporate a multimodal transformer to enhance high-level semantic feature fusion, further reducing cross-modal inconsistencies. Our extensive experiments across four widely used multimodal benchmarks demonstrate that DecAlign consistently outperforms state-of-the-art methods on five metrics. These results highlight the efficacy of DecAlign in improving cross-modal alignment and semantic consistency while preserving modality-unique features, marking a significant advancement in multimodal representation learning scenarios. Our project page is at https://taco-group.github.io/DecAlign/.

## 1 Introduction

Multimodal representation learning seeks to effectively integrate diverse modalities by capturing their shared semantics while retaining modality-unique characteristics. This goal has been pursued across numerous domains, including multimodal sentiment analysis (Lian et al., 2023; Das & Singh, 2023; Wang et al., 2024a), recommendation systems (Liu et al., 2024a; 2022), autonomous driving (Hwang et al., 2024; Xing et al., 2024b; Ma et al., 2025; Xing et al., 2024a), out-of-distribution detection (Dong et al., 2024; Li et al., 2024b), and general visual understanding and reasoning (Xing et al., 2025; Wang et al., 2024b). Despite significant advancements, the intrinsic heterogeneity among modalities—mainly due to divergent data distributions, various representation scales, and semantic granularities—remains a critical barrier that hampers effective cross-modal integration.

**Motivation.** This challenge is further intensified by the complex entanglement of modality-unique (heterogeneous) patterns and cross-modal common (homogeneous) semantics. Conventional multimodal fusion methods typically simplify the issue by projecting raw multimodal data into unified spaces via straightforward *concatenation* or *linear transformations*(Han et al., 2022; Zhang et al., 2023). However, this indiscriminate fusion often entangles modality-unique features with global shared semantics, leading to semantic interference, wherein detailed unimodal characteristics may disrupt global cross-modal relationships (Liang et al., 2024a; Xu et al., 2023). This phenomenon is particularly evident when dealing with dimensional mismatches, such as high-dimensional, spatially correlated image features paired with low-dimensional, temporally correlated text features (Wei et al., 2025; 2024; Zhu et al., 2024). These dimensional mismatches frequently lead to suboptimal alignment, causing either information redundancy or critical loss during fusion.

**Our Approach.** To overcome these limitations, we propose **DecAlign**, a hierarchical cross-modal alignment framework for multimodal representation learning. As illustrated in Figure 2, DecAlign

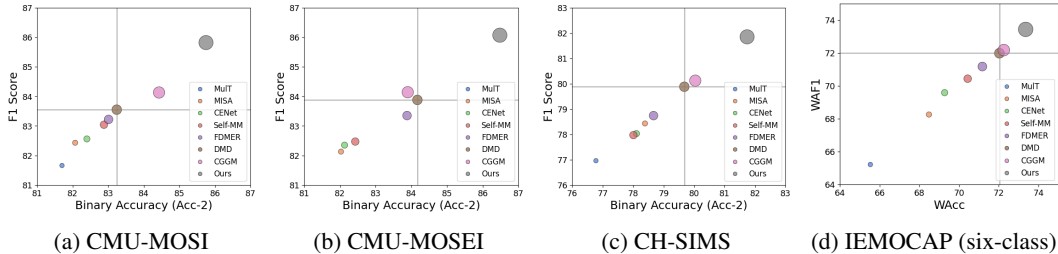

| (a) CMU-MOSI | (b) CMU-MOSEI | (c) CH-SIMS | (d) IEMOCAP (six-class) |

Figure 1: DecAlign achieves superior performance compared to state-of-the-art methods across multiple multimodal benchmarks. The bubble size represents relative model performance, illustrating the trade-off between Acc-2 and Binary F1 Score.

first explicitly decouples heterogeneous and homogeneous features through specialized encoders. Then, leveraging a dual-stream cross-modal alignment mechanism, DecAlign individually handles modality characteristics at different granularities: ❶ For heterogeneity, we propose **prototype-based optimal transport alignment** (Peyré & Cuturi, 2019) using Gaussian Mixture Modeling (GMM) (Bishop, 2006) and multi-marginal transport plans (Pass, 2015), effectively mitigating distribution discrepancies and constraining modality-unique interference. Additionally, we enhance semantic alignment and robustness through a multimodal transformer, which employs cross-modal attention mechanisms to bridge high-level semantic inconsistencies. ❷ For homogeneity, DecAlign achieves semantic consistency via **latent distribution matching** with Maximum Mean Discrepancy (MMD) regularization. Finally, we concatenate the aligned modality-unique features with modality-common features, passing them through a learnable projector for downstream tasks. Our key contributions are summarized as follows:

- **Modality Decoupling**. We propose DecAlign, a novel hierarchical cross-modal alignment framework that decouples multimodal features into modality-heterogeneous and modality-homogeneous components, allowing tailored strategies to capture both modality-unique characteristics and shared semantics.

- **Hierarchical Alignment Strategy**. We develop a dual-stream alignment mechanism that combines prototype-guided optimal transport and cross-modal transformers to handle modality heterogeneity, while applying latent space statistical matching to address homogeneity, substantially improving cross-modal semantic integration.

- **Empirical Evaluation**. Extensive experiments on four widely used benchmark datasets demonstrate that DecAlign consistently outperforms 13 state-of-the-art methods, validating its efficacy and generalizability for multimodal representation learning.

## 2 RELATED WORK (EXTENDED VERSION IN APPENDIX A)

**Multimodal Representation Learning.** This field integrates heterogeneous modalities into unified representations that capture complementary semantics (Qian et al., 2025; Liang et al., 2024b; Bayoudh, 2024; Wang et al., 2025). Advances include contrastive and masked modeling (Self-MM), and hierarchical graph contrastive learning (HGraph-CL) (Yu et al., 2021; Lin et al., 2022). Yet entanglement of heterogeneity and complementarity hampers leveraging both. To address this, MISA disentangles invariant and unique features, while DMD applies graph knowledge distillation (Hazarika et al., 2020; Li et al., 2023). However, global modeling dominates, often neglecting token-level inconsistencies. Our DecAlign introduces hierarchical alignment, moving from local to global, heterogeneity to homogeneity, for precise and consistent integration.

**Cross-Modal Alignment.** The core challenge in multimodal learning is structural, distributional, and semantic heterogeneity, which restricts feature synergy (Zhu et al., 2024). Main approaches include: ❶ Shared Representation. Learning a unified latent space for semantic consistency. CLIP aligns image–text pairs via large-scale contrastive learning (Radford et al., 2021; Gao et al., 2024), while Uni-Code uses disentangling and exponential moving average for stable alignment (Xia et al., 2024). ❷ Transformer-based Cross-Attention. Cross-attention dynamically captures information across modalities, as in multimodal transformers with disentangled or hierarchical fusion (Tsai et al., 2019; Yang et al., 2022; Hu et al., 2024). ❸ Modality Translation. Translation methods build

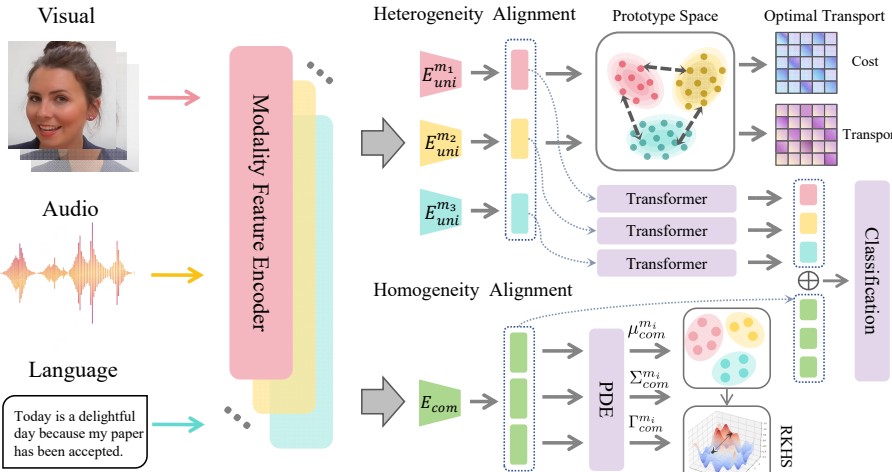

Figure 2: **The framework of our proposed DecAlign approach**, illustrated in a multimodal setting with visual, audio, and language inputs. Modality Feature Encoders first extract unimodal embeddings, which are then decoupled into modality heterogeneous and homogeneous components by modality-unique/common encoders. Heterogeneous features are aligned via optimal transport-based cross-modal prototypes, and homogeneous semantics are aligned through latent space semantics and Maximum Mean Discrepancy-based distribution matching. Heterogeneous features are refined by a multimodal transformer for capturing finer-grained cross-modal interactions, then concatenated with homogeneous features and passed through a fully connected layer for downstream tasks.

mappings through cross-modal generation or reconstruction, explicitly modeling dependencies (Liu et al., 2024b; Zeng et al., 2024; Tian et al., 2022). ❹ Knowledge Distillation. Distillation balances inter-modal contributions by transferring knowledge. DMD applies graph distillation for correlation modeling, and UMDF uses unified self-distillation for robust representation learning (Li et al., 2023; 2024a). Compared with methods that risk over-alignment and loss of modality-specific traits, our framework combines representation decoupling with hierarchical alignment to preserve unimodal uniqueness while ensuring semantic consistency.

## 3 METHOD

**Motivation and Overview**. The fundamental challenge in multimodal representation learning lies in effectively addressing the inherent conflicts between modality-unique characteristics and cross-modal semantic consistency. Two critical issues emerge: ❶ **Heterogeneity**: referring to inherent representation focus and distributional discrepancies among modalities that hinder cross-modal semantic alignment, ❷ **Homogeneity**: emphasizing the necessity of capturing shared semantics across modalities despite their inherent differences. To overcome these limitations, we propose DecAlign, a hierarchical cross-modal alignment framework that explicitly treats modality-unique and modality-common features with specific alignment strategies. As illustrated in Figure 2, DecAlign begins by decoupling multimodal representations into modality-unique (heterogeneous) and modality-common (homogeneous) features (Section 3.1). A hierarchical alignment mechanism is subsequently employed, combining prototype-guided multi-marginal optimal transport and cross-modal transformer for heterogeneous alignment (Section 3.2) and latent space semantic consistency with MMD regularization for homogeneous alignment (Section 3.3), ensuring the semantic consistency of modality-unique information and cross-modal commonality.

### 3.1 MULTIMODAL FEATURE DECOUPLING

Given a multimodal dataset with $M$ modalities, each modality $m$ provides features with its unique temporal length $T_m$ and feature dimension $d_m$. Due to this inherent variation across modalities, we apply modality-unique 1D temporal convolution layers that aggregate local temporal patterns and transform all features to the same temporal length $T_s$ and feature dimension $d_s$. The resulting unimodal features are expressed as: $\tilde{\mathbf{X}}_m \in \mathbb{R}^{T_s \times d_s}$. The primary challenge in multimodal tasks

lies in the inherent heterogeneity across modalities, hindering the integration of homogeneous features. To address this, we decouple the multimodal representations into **modality-common** features, which emphasize semantic consistency across modalities, and **modality-unique** features, capturing modality-unique characteristics with some redundancy. Building upon this, we employ three modality-unique encoders $\mathbf{E}_{\text{uni}}^{(m)}$ and a modality-shared encoder $\mathbf{E}_{\text{com}}$, to extract heterogeneous features as $\mathcal{F}_{\text{uni}}^{(m)} = \mathbf{E}_{\text{uni}}^{(m)}(\tilde{\mathbf{X}}_m)$ and cross-modal homogeneous features as $\mathcal{F}_{\text{com}}^{(m)} = \mathbf{E}_{\text{com}}(\tilde{\mathbf{X}}_m)$.

Considering the inherent heterogeneity and potential redundancy across modalities, we refine the decoupling process by explicitly separating modality-unique and modality-common features. All encoders are designed to produce representations with the same dimensionality to ensure compatibility. Instead of modeling distributions or computing mutual information which can be computationally expensive, we use cosine similarity to quantify their potential overlap. Hence, the loss of decoupling process is formally defined as:

$$\mathcal{L}_{dec} = \sum_{m=1}^{M} \frac{\mathcal{F}_{\text{uni}}^{(m)} \cdot (\mathcal{F}_{\text{com}}^{(m)})^{\text{T}}}{||\mathcal{F}_{\text{uni}}^{(m)}|| \, ||\mathcal{F}_{\text{com}}^{(m)}||} \tag{1}$$

## 3.2 HETEROGENEITY ALIGNMENT

In multimodal tasks, modality-unique features capture distinct characteristics specific to each modality. However, these features often differ significantly in spatial structure, scale, noise level, and density, making direct point-to-point alignment across modalities both unreliable and computationally expensive. Moreover, although these features vary in form, they frequently carry semantically aligned information when referring to the same underlying concept or object category. To effectively bridge modality-unique feature differences while preserving shared semantic structures, we introduce category prototypes as semantic anchors across modalities. These prototypes represent consistent semantic patterns underlying different modality-specific representations and serve as reference points to guide alignment. Building on this, we employ a prototype-guided multi-marginal optimal transport framework to achieve adaptive and fine-grained alignment across heterogeneous feature spaces.

**Prototype Generation.** To flexibly capture the complex distributions and potential correlations in multimodal data, we employ the Gaussian Mixture Model (GMM), which leverages its soft assignment mechanism and Gaussian distribution assumption to more accurately represent the prototype structures of different modality features. The GMM is fitted using the standard Expectation-Maximization algorithm, which iteratively estimates the mixture coefficients, means, and covariances to maximize the likelihood of the modality-unique features. We first model modality-unique features using GMM, with prototypes represented by the mean and covariance of Gaussian distributions:

$$\mathcal{P}_m = \{(\mu_m^1, \Sigma_m^1), (\mu_m^2, \Sigma_m^2), \ldots, (\mu_m^K, \Sigma_m^K)\} \tag{2}$$

where $K$ denotes the number of Gaussian components, which is set equal to the category number in downstream task, and $\mu_m^k, \Sigma_m^k$ represent the mean and covariance of the $k$-th Gaussian component for modality $m$, respectively. Then the probability of $n$-th sample $\mathbf{x}_n$ belonging to the $k$-th Gaussian component is calculated as:

$$w_m^n(k) = \frac{\pi_k \cdot \mathcal{N}(\mathbf{x}_m^n; \mu_m^k, \Sigma_m^k)}{\sum_{j=1}^{K} \pi_j \cdot \mathcal{N}(\mathbf{x}_m^i; \mu_m^j, \Sigma_m^j)} \tag{3}$$

$\pi_k$ is the mixture coefficient of the $k$-th Gaussian component, and $\mathcal{N}(\mathbf{x}_m^i; \mu_m^k, \Sigma_m^k)$ is the probability density function of the Gaussian distribution:

$$\mathcal{N}(\mathbf{x}_m^i; \mu_m^k, \Sigma_m^k) = \frac{\exp\left(-\frac{1}{2}(\mathbf{x}_m^i - \mu_m^k)^{\text{T}} \Sigma_m^{k-1}(\mathbf{x}_m^i - \mu_m^k)\right)}{(2\pi)^{d/2} |\Sigma_m^k|^{1/2}} \tag{4}$$

**Prototype-guided Optimal Transport.** The modality-unique features of different modalities often lie in distinct feature spaces with significant distributional differences, traditional point-to-point alignment methods struggle to capture both global and local relationships. To address this challenge in multimodal scenarios, we introduce a multi-marginal Optimal Transport approach to establish matches between distributions. The cross-modal prototype matching cost matrix is defined as:

$$C(k_1, k_2, \ldots, k_M) = \sum_{1 \le i \le j \le M} C_{i,j}(k_i, k_j) \tag{5}$$

where $C_{i,j}(k_i, k_j)$ represents the pairwise alignment cost between modalities $m_i$ and $m_j$:

$$C_{i,j}(k_i, k_j) = ||\mu_i^{k_i} - \mu_j^{k_j}||^2 + \text{Tr}(\Sigma_i^{k_i} + \Sigma_j^{k_j} - 2(\Sigma_i^{k_i}\Sigma_j^{k_j})^{\frac{1}{2}}) \quad (6)$$

The optimization objective for cross-modal prototype alignment aims to minimize the total alignment cost across all modalities while satisfying marginal distribution constraints. The objective function is

$$T^* = \arg\min_T \sum_k T(k) \cdot C(k) + \lambda \sum_k T(k) \log T(k), \quad (7)$$

where $k \in \{k_1, k_2, \ldots, k_M\}$ denotes the set of indices spanning all prototype combinations across the $M$ modalities, $T(k)$ represents the joint transportation matrix, and $C(k)$ is the joint cost matrix. The second term introduces entropy regularization to promote smoother and more robust solutions. The transport plan matrix $T(k)$ is further constrained to ensure consistency across modalities, satisfying the following marginal distribution constraints:

$$\sum_{k_j:j\neq i} T(k_1, k_2, \ldots, k_M) = \nu_i(k_i), \forall i \in \{1, 2, \ldots, M\}, \forall k_i, \quad (8)$$

where $\nu_i(j_i)$ represents the marginal distribution of modality $m_i$ over its prototypes. Combining global alignment via Optimal Transport and local alignment through sample-to-prototype calibration, the overall heterogeneity alignment loss is defined as:

$$\mathcal{L}_{hete} = \sum_k T^*(k) \cdot C(k) + \frac{1}{N} \sum_{n=1}^N \sum_{k=1}^K w_i^n(k) \cdot ||\mathcal{F}_i^n - \mu_{j\neq i}^k||^2. \quad (9)$$

The first term, $\mathcal{L}_{OT}$, aligns the distributions of prototypes across modalities, ensuring global consistency. The second term $\mathcal{L}_{Proto}$ ensures fine-grained alignment by minimizing the weighted distance between samples $x_i^n$ in source modality $i$ and prototypes in target modality $j$. By combining $\mathcal{L}_{OT}$ and $\mathcal{L}_{Proto}$, this heterogeneous alignment loss captures both global and local relationships, providing a robust mechanism for aligning heterogeneous modalities in a unified feature space.

### 3.3 HOMOGENEITY ALIGNMENT

While different modalities exhibit unique characteristics in their representations, they also share common elements that convey the same semantic information. To effectively uncover and align these shared features, it is crucial to address the inherent challenges posed by modality-unique variations and residual inconsistencies in their distributions.

**Latent Space Semantic Alignment.** To address the global offset and semantic inconsistencies in modality-common features and mitigate information distortion during feature fusion, we model modality feature distributions using Gaussian distributions. By mapping representations into a latent space, we quantify differences in position, shape, and symmetry through mean, covariance, and skewness, where skewness is further incorporated to capture asymmetry in the distribution of modality-common features, enabling the alignment to account for non-Gaussian semantic variations and improve cross-modal consistency. Specifically, for modality-common features, their distributions are approximated as $\mathcal{Z}_{com}^{m_i} \sim \mathcal{N}(\mu_{com}^{m_i}, \Sigma_{com}^{m_i}, \Gamma_{com}^{m_i})$, where $\mu_{com}^{m_i}$, $\Sigma_{com}^{m_i}$ and $\Gamma_{com}^{m_i}$ represent the mean, covariance and skewness of the common features for modality $m_i$, respectively. Their detailed formulas are discussed in the Appendix B.6. To ensure semantic consistency across modalities, we define the latent space semantic alignment loss as:

$$\mathcal{L}_{sem} = \frac{1}{M(M-1)} \sum_{1 \leq i < j \leq M} \left( ||\mu_{com}^{m_i} - \mu_{com}^{m_j}||^2 + ||\Sigma_{com}^{m_i} - \Sigma_{com}^{m_j}||_F^2 + ||\Gamma_{com}^{m_i} - \Gamma_{com}^{m_j}||^2 \right). \quad (10)$$

**Cross-Modal Distribution Alignment.** To flexibly model the latent distribution space of modality-homogeneous features extracted by the shared encoder without relying on prior knowledge, we use Probabilistic Distribution Encoder (PDE) to encode feature distributions in latent space. PDE outputs are compared across modalities using the Maximum Mean Discrepancy (MMD) metric, which evaluates the distance between distributions by mapping them into a Reproducing Kernel Hilbert Space (RKHS) and measuring the difference between their mean embeddings. This kernel-based

formulation enables non-parametric modeling and captures higher-order statistical properties in a unified space. The discrepancy of cross-modal distribution is then quantified as:

$$\mathcal{L}_{\text{MMD}} = \frac{2}{M(M-1)} \sum_{1 \leq i < j \leq M} \Big[ \mathbb{E}_{x,x' \sim \mathcal{Z}_{com}^{m_i}}[k(x,x')]$$

$$+ \mathbb{E}_{y,y' \sim \mathcal{Z}_{com}^{m_j}}[k(y,y')] - 2\,\mathbb{E}_{x \sim \mathcal{Z}_{com}^{m_i}, y \sim \mathcal{Z}_{com}^{m_j}}[k(x,y)] \Big] \tag{11}$$

where $k(\cdot, \cdot)$ is the Gaussian kernel function defined with its kernel bandwidth parameter $\sigma$:

$$k(x,y) = \exp\Big(-\frac{||x-y||^2}{2\sigma^2}\Big) \tag{12}$$

By conducting latent space semantic alignment followed by MMD-based distribution correction, we establish a hierarchical homogeneity alignment mechanism that effectively achieves semantic and distributional consistency of modality-common features. The overall loss for homogeneity alignment is $\mathcal{L}_{homo} = \mathcal{L}_{sem} + \mathcal{L}_{\text{MMD}}$.

### 3.4 MULTIMODAL FUSION AND PREDICTION

Recognizing the unique characteristics of multimodal heterogeneous representations—such as syntactic structures in language, spatial layouts in vision, and temporal patterns in audio—we incorporate modality-specific transformers (Tsai et al., 2019) to enhance global temporal and contextual modeling. While prior alignment places modality-unique features in semantically consistent spaces, these representations still contain rich intra-modal information that benefits from further refinement. Using separate transformers per modality does not undermine alignment, as the representation space has been regularized by alignment losses. Instead, the transformers serve as modality-aware refiners. Their outputs are concatenated with modality-common features, enabling both shared semantics and modality-specific cues to jointly inform the final prediction, which is generated by a fully connected layer. The overall optimization objective of our framework is defined as:

$$\mathcal{L}_{total} = \mathcal{L}_{task} + \mathcal{L}_{dec} + \alpha \mathcal{L}_{hete} + \beta \mathcal{L}_{homo} \tag{13}$$

where $\mathcal{L}_{task}$ represents the task-specific loss, such as cross-entropy for classification tasks or mean squared error for regression. $\alpha$ and $\beta$ are trade-off hyperparameters for the losses of heterogeneous and homogeneous alignment, with their sensitivity analyzed in Section 4.3.

## 4 EXPERIMENTS

**Dataset and Metric Description.** We evaluate DecAlign on four common multimodal datasets: CMU-MOSI (Zadeh et al., 2016), CMU-MOSEI (Zadeh et al., 2018), CH-SIMS (Yu et al., 2020) and IEMOCAP (Busso et al., 2008). For CMU-MOSI and CMU-MOSEI, following prior works (Liang et al., 2021; Li et al., 2023; Zhou et al., 2025), we evaluate performance using binary accuracy (Acc-2), 7-class accuracy (Acc-7), and Binary F1 Score. Acc-2 reflects whether the a sample is predicted as negative, while sentiment intensity prediction is further assessed via Mean Absolute Error (MAE) and Pearson Correlation (Corr) to capture deviation and linearity. For CH-SIMS, we adopt MAE and F1 Score. IEMOCAP follows (Lian et al., 2023; Fu et al., 2024; Zhang et al., 2024) with weighted accuracy (WAcc) and weighted average F1 Score (WAF1), accounting for class distribution imbalance. Detailed dataset and metric descriptions are provided in Appendix B.

**Implementation Details.** Consistent with previous studies (Li et al., 2023; Wang et al., 2023), we use the MMSA-FET Toolkit (Yu et al., 2021) for feature extraction on all datasets except IEMOCAP, for which we follow the pre-processing procedure described in prior representative work (Lian et al., 2023). We train DecAlign for 50 epochs using Adam optimizer with a batch size of 32 on an NVIDIA A6000. Further details regarding hyperparameter settings are provided in Appendix B.3, and feature extraction is described in Appendix B.4.

### 4.1 COMPARISON ANALYSIS (EXTENDED VERSION IN APPENDIX C)

We compare DecAlign with a range of state-of-the-art methods under a unified experimental environment and consistent dataset splits. These baselines include MFM (Tsai et al., 2018), MulT (Tsai et al.,

| Models | CMU-MOSI | | | CMU-MOSEI | | | IEMOCAP (six-class) | | CH-SIMS | |
|---|---|---|---|---|---|---|---|---|---|---|
| | MAE (↓) | Acc-2 (↑) | F1 Score (↑) | MAE (↓) | Acc-2 (↑) | F1 Score (↑) | WAcc (↑) | WAF1 (↑) | MAE (↓) | F1 Score (↑) |
| MFM (Tsai et al., 2018) | 0.951 | 78.18 | 78.10 | 0.681 | 78.93 | 76.45 | 63.38 | 63.41 | 0.471 | 75.28 |
| MulT (Tsai et al., 2019) | 0.846 | 81.70 | 81.66 | 0.673 | 80.85 | 80.86 | 65.53 | 65.21 | 0.455 | 76.96 |
| PMR (Fan et al., 2023) | 0.895 | 79.88 | 79.83 | 0.645 | 81.57 | 81.56 | 67.04 | 67.01 | 0.445 | 76.55 |
| CubeMLP (Sun et al., 2022) | 0.838 | 81.85 | 81.74 | 0.601 | 81.36 | 81.75 | 66.43 | 66.41 | 0.459 | 77.85 |
| MUTA-Net (Tang et al., 2023) | 0.767 | 82.12 | 82.07 | 0.617 | 81.76 | 82.01 | 67.44 | 68.78 | 0.443 | 77.21 |
| MISA (Hazarika et al., 2020) | 0.788 | 82.07 | 82.43 | 0.594 | 82.03 | 82.13 | 68.48 | 68.25 | 0.437 | 78.43 |
| CENet (Wang et al., 2022) | 0.745 | 82.40 | 82.56 | 0.588 | 82.13 | 82.35 | 69.27 | 69.58 | 0.454 | 78.03 |
| Self-MM (Yu et al., 2021) | 0.765 | 82.88 | 83.04 | 0.576 | 82.43 | 82.47 | 70.35 | 70.43 | 0.432 | 77.97 |
| FDMER (Yang et al., 2022) | 0.760 | 83.01 | 83.22 | 0.571 | 83.88 | 83.35 | 71.33 | 71.17 | 0.424 | 78.74 |
| AOBERT (Kim & Park, 2023) | 0.780 | 83.03 | 83.02 | 0.588 | 83.90 | 83.64 | 71.04 | 70.89 | 0.430 | 78.55 |
| DMD (Li et al., 2023) | 0.744 | 83.24 | 83.55 | 0.561 | 84.17 | 83.88 | 72.03 | 71.98 | 0.421 | 79.88 |
| ReconBoost (Hua et al., 2024) | 0.793 | 82.59 | 82.72 | 0.599 | 82.98 | 83.14 | 71.44 | 71.58 | 0.413 | 80.41 |
| CGGM (Guo et al., 2025) | 0.787 | 82.73 | 82.89 | 0.584 | 83.72 | 83.94 | 72.25 | 72.17 | 0.417 | 80.12 |
| DecAlign (Ours) | **0.735** | **85.75** | **85.82** | **0.543** | **86.48** | **86.07** | **73.35** | **73.43** | **0.403** | **81.85** |

Table 1: Performance comparison across four widely used datasets under a unified experimental setting with consistent data splits to ensure a fair evaluation. Symbols ↑ and ↓ indicate that higher or lower values are better, respectively. Best results are highlighted in **bold**, and second-best results are underlined. All reported results are averaged over **five** runs on the test set.

2019), PMR (Fan et al., 2023), CubeMLP (Sun et al., 2022), MUTA-Net (Tang et al., 2023), MISA (Hazarika et al., 2020), CENet (Wang et al., 2022), Self-MM (Yu et al., 2021), FDMER (Yang et al., 2022), AOBERT (Kim & Park, 2023), DMD (Li et al., 2023), ReconBoost (Hua et al., 2024), and CGGM (Guo et al., 2025). Table 1, 5, 6, 7, along with Figure 1, present a comprehensive comparison of our DecAlign framework against 13 state-of-the-art methods on four widely used datasets. To account for statistical significance and reduce the influence of randomness, the reported performance of DecAlign is averaged over **five** independent runs. The comparison reveals that DecAlign exhibits a stronger ability to capture subtle variations in continuous target values, as well as a more precise distinction among discrete categories. Its consistent performance across diverse datasets indicates an enhanced capacity for modeling both continuous and categorical patterns within multimodal data, reflecting a more comprehensive understanding of complex cross-modal interactions.

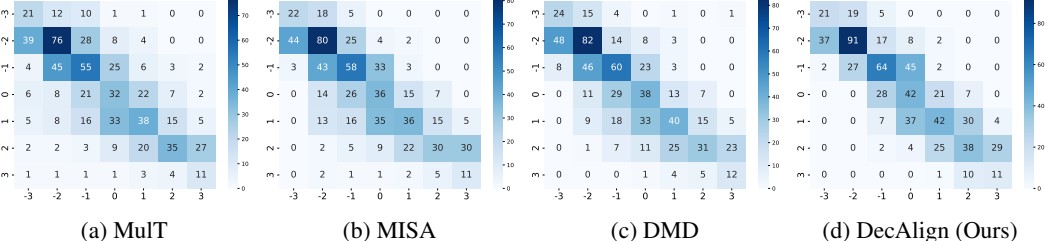

|  (a) MulT | (b) MISA | (c) DMD | (d) DecAlign (Ours) |

Figure 3: Comparison of predicted versus ground truth category distributions for four representative models on the CMU-MOSI dataset.

**Transformer-based methods.** Compared to Transformer-based methods such as MulT (Tsai et al., 2019), Self-MM (Yu et al., 2021), PMR (Fan et al., 2023), and MUTA-Net (Tang et al., 2023), which rely on cross-attention mechanism for global feature fusion, DecAlign overcomes modality-unique interference and local semantic inconsistencies. Transformer-based models assume a shared latent space, often causing dominant modalities to overshadow weaker ones, leading to information loss. In contrast, DecAlign explicitly disentangles modality-heterogeneous and modality-homogeneous features, leveraging prototype-based optimal transport for fine-grained alignment and latent space semantic alignment with MMD regularization for global consistency. This mitigates modality interference, reducing MAE and improving Corr, while enhancing classification performance.

**Feature Decoupling-based methods.** While multimodal feature decoupling methods such as MISA (Hazarika et al., 2020), FDMER (Yang et al., 2022), and DMD (Li et al., 2023) alleviate modality interference, they primarily focus on global alignment, often overlooking token-level inconsistencies. This limitation hinders fine-grained multimodal integration, particularly in tasks requiring precise semantic fusion. DecAlign overcomes this challenge through a dual-stream hierarchical alignment strategy, integrating prototype-based transport for local alignment with semantic consistency constraints for robust global integration. This enables more expressive multimodal representations, leading to superior performance across both regression and classification metrics.

**Confusion Matrix Analysis.** To further demonstrate the superiority of our performance and validate the effectiveness of our proposed approach, we analyze the confusion matrix of DecAlign in comparison with representative works in the field of multimodal sentiment analysis, including MulT

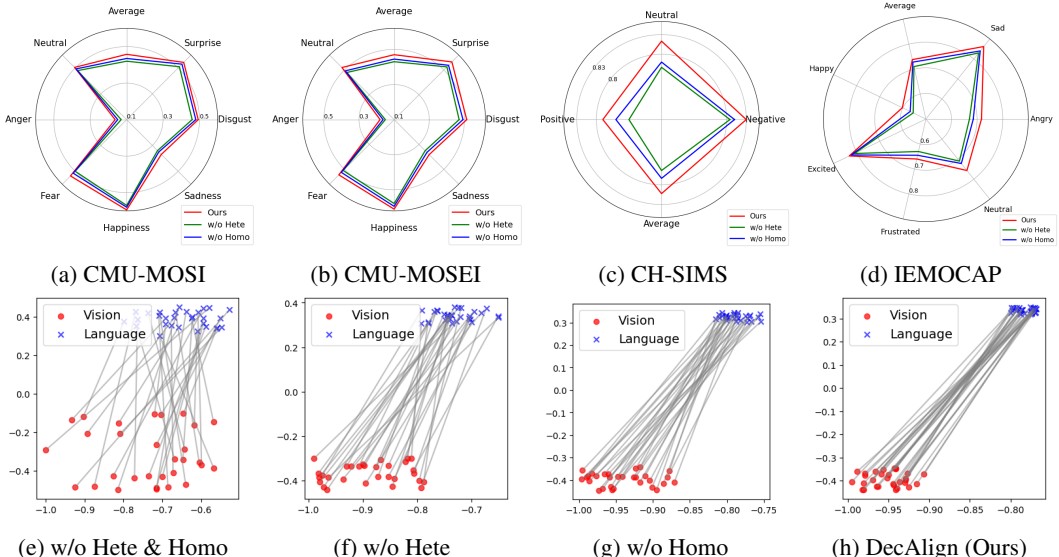

Figure 4: Visualization of Ablation Studies. (a)–(d) illustrate the performance comparison across different emotion categories on four benchmarks, (e)–(h) visualize the modality gap between visual and language modalities on the CMU-MOSEI dataset.

| Key Modules | | | CMU-MOSI | | CMU-MOSEI | | Alignment Strategies | | | | CMU-MOSI | | CMU-MOSEI | |
|---|---|---|---|---|---|---|---|---|---|---|---|---|---|---|
| MFD | Hete | Homo | MAE | F1 | MAE | F1 | Proto-OT | CT | Sem | MMD | MAE | F1 | MAE | F1 |
| ✓ | ✓ | ✗ | 0.747 | 84.46 | 0.562 | 84.74 | ✓ | ✓ | ✓ | ✗ | 0.741 | 84.61 | 0.564 | 85.26 |
| ✓ | ✗ | ✓ | 0.754 | 84.03 | 0.588 | 84.37 | ✓ | ✓ | ✗ | ✓ | 0.738 | 84.73 | 0.553 | 85.33 |
| ✓ | ✗ | ✗ | 0.784 | 81.92 | 0.632 | 82.22 | ✓ | ✗ | ✓ | ✓ | 0.743 | 84.36 | 0.619 | 85.21 |
| ✗ | ✗ | ✗ | 0.794 | 81.56 | 0.624 | 81.87 | ✗ | ✓ | ✓ | ✓ | 0.748 | 84.17 | 0.624 | 85.03 |

Table 2: Ablation study on key modules (left) and alignment strategies (right) for CMU-MOSI and CMU-MOSEI datasets.

(Tsai et al., 2019), MISA (Hazarika et al., 2020), and DMD (Li et al., 2023). As shown in Figure 3, DecAlign achieves a more balanced and accurate sentiment classification across different sentiment intensity levels, significantly reducing misidentification errors, particularly in distinguishing subtle sentiment variations.

Compared to other methods, DecAlign exhibits stronger diagonal dominance, reflecting higher sentiment classification accuracy. Notably, in extreme sentiment classes (-3 and +3), where existing models often misclassify samples, DecAlign significantly reduces confusion with adjacent sentiment levels. The higher concentration of correctly predicted samples in moderate sentiment categories (-1, 0, and 1) further demonstrates its ability to capture fine-grained sentiment distinctions, mitigating bias toward neutral or extreme labels. Furthermore, unlike MulT, MISA, and DMD, which struggle with negative-to-neutral misidentification, DecAlign achieves clearer separation between sentiment classes, ensuring more robust and interpretable predictions. This improvement is particularly evident in -2 and +2 classes, where DecAlign minimizes misidentification into adjacent categories, validating the effectiveness of its hierarchical alignment strategy in capturing both modality-unique nuances and shared semantic patterns.

## 4.2 ABLATION STUDIES (EXTENDED VERSION IN APPENDIX C.3)

To further evaluate the contributions of individual components in DecAlign, we conduct ablation studies on the MOSI and MOSEI dataset, while results on other benchmarks are given in the Appendix. The first study examines the impact of key model components, while the second focuses on the effectiveness of specific alignment strategies.

**Impact of key components.** We evaluate the impact of Multimodal Feature Decoupling (MFD), Heterogeneous (Hete), and Homogeneous (Homo) Alignment on model performance using MAE and Binary F1 Score (Table 2). The full model achieves the best results, confirming the significance of hierarchical alignment. Removing Homogeneous Alignment slightly increases MAE and lowers Acc-2, indicating the importance of intra-modal consistency. Eliminating Heterogeneous Alignment

leads to a greater drop, showing that modality-unique interference affects feature integration. The absence of both alignments causes substantial performance degradation, highlighting the need to disentangle modality-homogeneous and modality-heterogeneous features.

Additionally, Figure 4 (a)-(d) visualizes the ablation results across different sentiment categories, illustrating the performance variations when heterogeneous and homogeneous alignment modules are frozen. The degradation across sentiment categories further validates the necessity of a hierarchical alignment strategy to maintain robust performance across diverse emotional expressions. Notably, even when any single alignment module is disabled, the F1 Score remains higher than many state-of-the-art methods, including FDMER, AOBERT, and DMD, demonstrating the effectiveness of our proposed alignment approach from both heterogeneous and homogeneous perspectives. The most severe performance degradation occurs when MFD is removed, demonstrating that preserving modality-unique information before fusion is crucial. This underscores the effectiveness of integrating heterogeneous and homogeneous representations for better sentiment analysis.

**Impact of specific alignment strategies.** We further evaluate the contribution of Prototype-Based Optimal Transport (Proto-OT), Cross-model Transformer (CT), Semantic Consistency (Sem), and Maximum Mean Discrepancy (MMD) Regularization to DecAlign's performance, as shown in Table 2. Removing MMD regularization leads to a slight performance drop, highlighting its role in global latent space alignment and feature coherence. The exclusion of semantic consistency further degrades performance, indicating that enforcing semantic alignment enhances multimodal feature integration. The most substantial drop occurs when contrastive training is removed, showing its critical role in learning discriminative multimodal representations. Similarly, eliminating Proto-OT results in a notable decline in both regression and classification metrics, demonstrating that fine-grained alignment through optimal transport significantly improves multimodal collaborative prediction performance.

**Analysis of modality Gap.** Figure 4 (e)-(h) presents a case study on vision and language modalities, demonstrating how DecAlign mitigates the modality gap to enhance alignment. Models without heterogeneous or homogeneous alignment exhibit significantly larger gaps, hindering cross-modal fusion. These results further validate the effectiveness of our hierarchical alignment strategy. Extended Analysis will be shown in Appendix C.4.

## 4.3 PARAMETER SENSITIVITY ANALYSIS

To analyze the impact of hyper-parameters $\alpha$ and $\beta$ on DecAlign, we conduct an extensive grid search and evaluate the model's Binary F1 Score across different parameter settings on MOSI and MOSEI datasets. Figure 5 presents a heatmap visualization of the results, where darker shades indicate higher performance. The optimal setting is $\alpha = 0.05, \beta = 0.05$, achieving the highest Performance across both datasets. Larger values cause a sharp performance drop, indicating that excessive alignment constraints hinder effective fusion. Smaller $\alpha$ values with moderate $\beta$ yield strong performance, highlighting the importance of balancing prototype-based alignment and semantic consistency for optimal multimodal learning.

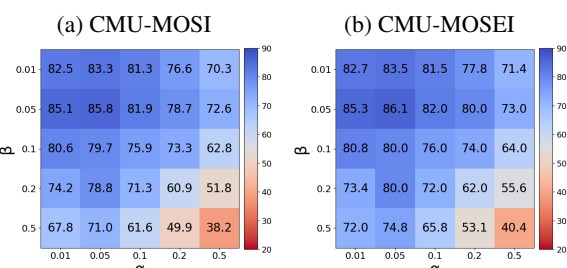

Figure 5: Hyperparameter sensitivity analysis on CMU-MOSI and CMU-MOSEI in terms of Binary F1 Score.

## 5 CONCLUSION

We present DecAlign, a hierarchical framework for decoupled multimodal representation learning that separately aligns modality-unique and modality-common features. Through prototype-guided optimal transport and latent semantic alignment, our method captures both global distributions and local semantics across modalities. Experiments on multiple benchmarks validate its effectiveness.

ETHICS STATEMENT

This work uses only publicly available benchmark datasets (CMU-MOSI, CMU-MOSEI, CH-SIMS, and IEMOCAP) under their respective licenses. No new human data was collected, and all experiments report only aggregated results without attempting to identify individuals. We caution against potential misuse of multimodal sentiment analysis for surveillance or profiling and release our code solely for research purposes.

REPRODUCIBILITY STATEMENT

We ensure reproducibility of DecAlign through transparent datasets, model configurations, and released code.

❶ **Datasets.** Experiments are conducted on four public benchmarks: CMU-MOSI, CMU-MOSEI, CH-SIMS, and IEMOCAP (six-class). Standard official splits are used for all datasets; statistics and task setups are given in Appendix B.

❷ **Feature Extraction.** For MOSI/MOSEI/CH-SIMS, we employ MMSA-FET (Yu et al., 2021); for IEMOCAP, we follow the pipeline of Lian et al. (2023). Text features come from `bert-base-uncased` (English) or `bert-base-chinese` (CH-SIMS); visual and acoustic features are extracted via OpenFace and COVAREP, with details in Appendix B.4.

❸ **Model and Training.** DecAlign employs a unified backbone across datasets, consisting of four Transformer layers and Conv1D kernels of size 5 for language, audio, and visual streams. The DST feature dimensions and attention heads are tuned according to dataset scale (e.g., [32, 8] for MOSI/CH-SIMS, [64, 8] for MOSEI, [48, 4] for IEMOCAP), as summarized in Table 4. Training is conducted for 50 epochs with the Adam optimizer using a batch size of 32, weight decay of 0.005, and scheduler patience of 5 on a single NVIDIA A6000 GPU. Learning rates are dataset-specific: $5 \times 10^{-5}$ for MOSI and CH-SIMS, and $1 \times 10^{-4}$ for MOSEI and IEMOCAP. Gradient clipping thresholds are set between 0.5 and 0.6 depending on the dataset to stabilize optimization. To ensure faithful reproduction, Appendix B.3 provides a complete list of dataset-specific hyperparameter configurations, including all dropout ratios, optimization schedules, and pretrained backbones.

❹ **Evaluation.** We adopt standard metrics: MAE, Corr, Acc-2/Acc-7, and F1 for MOSI/MOSEI; MAE, Corr, Acc-3, and F1 for CH-SIMS; WAcc and WAF1 for IEMOCAP (Appendix B.5). Results are averaged over five runs with fixed random seeds $\{1, 2, 3, 4, 5\}$.

❺ **Ablations and Analysis.** Ablation experiments remove individual modules (MFD, Hete, Homo) or alignment strategies (Proto-OT, CT, Sem, MMD), with results reported in Table 2. Figure 4 visualizes modality gaps under ablation settings. These analyses confirm the complementary roles of hierarchical alignment strategies.

❻ **Hyperparameter Settings.** To ensure faithful reproduction, we provide complete dataset-specific hyperparameter configurations in Appendix B.3. These include all dropout ratios across modalities (attention, embedding, residual, ReLU, and output), text-specific dropout, and learning rate schedules. We also detail the DST feature dimensions and number of heads for each dataset, Conv1D kernel sizes for language/audio/visual streams, and the number of Transformer layers. Optimization parameters such as batch size, learning rate, weight decay, gradient clipping thresholds, and scheduler patience are explicitly listed. Finally, we specify the pretrained model backbones used (`bert-base-uncased` for English datasets and `bert-base-chinese` for CH-SIMS). By consolidating these hyperparameters in a single appendix table (Table 4), we provide a transparent and comprehensive reference that enables researchers to reproduce our reported results without ambiguity.

❼ **Code Release.** We provide training and evaluation codes, dataset-specific configs, feature extraction pipelines, pretrained checkpoints, logs, and visualization scripts. Together with Appendices B–C.4, these artifacts enable faithful end-to-end reproduction of all reported results.

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

APPENDIX

In this appendix, we present additional related work, dataset descriptions, hyperparameter settings, experimental setup, comprehensive evaluation results, and extended experimental analyses. The detailed contents are organized as follows:

## A RELATED WORKS

### A.1 MULTIMODAL REPRESENTATION LEARNING

Multimodal representation learning aims to integrate heterogeneous data from diverse modalities into a cohesive framework that captures complementary semantic information (Qian et al., 2025; Liang et al., 2024b; Bayoudh, 2024; Wang et al., 2025). Recent methods have achieved significantly performance improvements by leveraging representation-based and cross-modal interaction approaches. Specifically, Self-MM (Yu et al., 2021) applies self-supervised contrastive learning and masked modeling to enhance the mutual information across modalities, HGraph-CL (Lin et al., 2022) introduces hierarchical graph contrastive learning to model intricate interactions across modalities. However, the heterogeneity and complementary information inherent in multimodal representations are intrinsically entangled, making it challenging to fully harness their complementary strengths while preserving their unique characteristics. Inspired by this insight, MISA (Hazarika et al., 2020) separates multimodal representations into modality-invariant and modality-unique features with contrastive and reconstruction losses, DMD (Li et al., 2023) further introduces graph cross-modal knowledge distillation to explicitly model the correlations across modalities. However, existing methods are often constrained to modeling modalities from a global perspective, overlooking the token-level local semantic inconsistencies that arise in cross-modal interactions. Our proposed DecAlign enables fine-grained multimodal representation learning through hierarchical alignment, progressing from local to global and heterogeneity to homogeneity, ensuring precise cross-modal integration and semantic consistency.

### A.2 CROSS-MODAL ALIGNMENT

The core challenge in multimodal tasks lies in the inherent heterogeneity across modalities (Zhu et al., 2024), characterized by structural, distributional, and semantic disparities, which restricts the

effective synergy of multimodal homogeneous features. To address this, existing solutions can be broadly categorized as follows: ❶ Shared Representation, which aims to learn a unified latent space for cross-modal semantic consistency. For example, CLIP-based methods (Radford et al., 2021; Gao et al., 2024) use contrastive learning to align image-text pairs in a shared embedding space, while Uni-Code (Xia et al., 2024) employs cross-modal information disentangling and exponential moving average to align semantically equivalent information in a shared latent space. ❷ Transformer-based methods that apply cross-attention to dynamically capture key information in cross-modal interactions (Tsai et al., 2019; Yang et al., 2022; Hu et al., 2024). ❸ Modality Translation, which establishes mappings between modalities through cross-modal generation or reconstruction (Liu et al., 2024b; Zeng et al., 2024; Tian et al., 2022). ❹ Cross-Modal Knowledge Distillation, which addresses inter-modal contribution imbalances and explores cross-modal correlations. For example, DMD (Li et al., 2023) employs graph distillation for dynamic knowledge transfer, and UMDF (Li et al., 2024a) uses unified self-distillation to learn robust representations from consistent multimodal distributions. Unlike methods that overemphasize homogeneous information, we tackle the issue of over-alignment diminishing modality-unique characteristics through representation decoupling and a hierarchical alignment mechanism, ensuring cross-modal semantic consistency while retaining unimodal characteristics.

## B  EXPERIMENTAL DETAILS

### B.1  MOTIVATION FOR DATASET SELECTION

To comprehensively evaluate DecAlign's effectiveness in multimodal sentiment analysis, we select four widely used benchmarking datasets. CMU-MOSI and CMU-MOSEI are well-established English multimodal sentiment datasets, enabling a direct and fair comparison with prior approaches. CH-SIMS extends our evaluation to a Chinese dataset, ensuring cross-linguistic generalization. Additionally, IEMOCAP (six-class) is included to assess the model's performance in fine-grained emotion classification. By conducting experiments across datasets with different languages, sentiment labels, and scales, we provide a thorough assessment of DecAlign's robustness and applicability.

| Dataset | # Train | # Test | # Category | Modality | | |
| --- | --- | --- | --- | --- | --- | --- |
| | | | | Audio | Visual | Text |
| CMU-MOSEI | 16327 | 4659 | 2 & 7 | ✓ | ✓ | ✓ |
| CMU-MOSI | 1284 | 686 | 2 & 7 | ✓ | ✓ | ✓ |
| CH-SIMS | 1368 | 457 | 3 | ✓ | ✓ | ✓ |
| IEMOCAP | 5810 | 1623 | 6 | ✓ | ✓ | ✓ |

Table 3: Statistical information on four chosen datasets.

### B.2  DETAILED DATASET DESCRIPTION

**CMU-MOSI** consists of 2,199 monologue movie review clips, each annotated with a sentiment score ranging from -3 (highly negative) to +3 (highly positive). It contains word-aligned multimodal signals, including textual, visual, and acoustic features. CMU-MOSI is commonly used for both sentiment classification and regression tasks, making it a crucial benchmark for evaluating multimodal models.

**CMU-MOSEI** extends CMU-MOSI by providing a significantly larger dataset with 22,856 opinion-based clips covering diverse topics, speakers, and recording conditions. Similar to CMU-MOSI, it includes multimodal data aligned at the word level and sentiment scores in the range of -3 to +3. Due to its large-scale and diverse nature, CMU-MOSEI is used to assess model generalization across various domains.

**CH-SIMS** consists of 38,280 Chinese utterances designed for multimodal sentiment analysis in Mandarin. Each sample includes textual, visual, and acoustic information, with sentiment labels ranging from -1 (negative) to +1 (positive). CH-SIMS enables research in cross-lingual sentiment analysis and serves as an important benchmark for multimodal sentiment models in Chinese contexts.

**IEMOCAP (six-class).** The IEMOCAP dataset comprises 10,039 dynamic utterances annotated with six emotion categories: angry, happy, sad, neutral, excited, and frustrated. Each sample contains textual, visual, and acoustic modalities. Due to its imbalanced class distribution, weighted accuracy (WAcc) and weighted average F1 score (WAF1) are commonly adopted to ensure fair performance evaluation in emotion recognition tasks.

### B.3 HYPER-PARAMETER SETTINGS

Table 4 summarizes the dataset-specific hyperparameter configurations used in DECALIGN. We observe that while most parameters are kept consistent across datasets, several key components are tuned to accommodate dataset characteristics. For example, the dropout rates applied to different modalities (audio, visual, and text) vary slightly: MOSI and CH-SIMS adopt relatively higher dropout for text and output layers (0.4–0.6), reflecting their smaller dataset sizes and the need for stronger regularization, whereas MOSEI and IEMOCAP use more moderate dropout values (0.2) to balance regularization and information retention.

In terms of feature representation, the DST feature dimension and attention heads are adjusted according to dataset scale: MOSEI employs the largest dimension setting ([64, 8]), while IEMOCAP uses a smaller but deeper configuration ([48, 4]). Other architectural choices, such as Conv1D kernel size, transformer depth (4 layers), and batch size (32), remain uniform across all datasets to ensure comparability.

For optimization, the learning rate is tuned between $5 \times 10^{-5}$ (MOSI and CH-SIMS) and $1 \times 10^{-4}$ (MOSEI and IEMOCAP), with weight decay fixed at 0.005 and gradient clipping set between 0.5 and 0.6. The scheduler patience is consistently set to 5 across datasets. Regarding pretrained models, English datasets (MOSI, MOSEI, IEMOCAP) utilize `bert-base-uncased`, while the Chinese dataset CH-SIMS employs `bert-base-chinese`.

Table 4: Dataset-specific Hyperparameter Settings for DecAlign

| Hyperparameter | MOSI | MOSEI | IEMOCAP | CH-SIMS |
|---|---|---|---|---|
| Attention Dropout (Audio) | 0.3 | 0.2 | 0.2 | 0.3 |
| Attention Dropout (Visual) | 0.1 | 0.2 | 0.2 | 0.1 |
| Attention Dropout (Text) | 0.4 | 0.2 | 0.2 | 0.4 |
| ReLU Dropout | 0.1 | 0.2 | 0.2 | 0.1 |
| Embedding Dropout | 0.3 | 0.2 | 0.2 | 0.3 |
| Residual Dropout | 0.1 | 0.2 | 0.2 | 0.1 |
| Output Dropout | 0.6 | 0.2 | 0.2 | 0.6 |
| Text Dropout | 0.5 | 0.2 | 0.2 | 0.5 |
| DST Feature Dim / Heads | [32, 8] | [64, 8] | [48, 4] | [32, 8] |
| Conv1D Kernel Size (L/A/V) | 5 / 5 / 5 | 5 / 5 / 5 | 5 / 5 / 5 | 5 / 5 / 5 |
| Transformer Levels (nlevels) | 4 | 4 | 4 | 4 |
| Batch Size | 32 | 32 | 32 | 32 |
| Learning Rate | 5e-5 | 1e-4 | 1e-4 | 5e-5 |
| Weight Decay | 0.005 | 0.005 | 0.005 | 0.005 |
| Gradient Clipping | 0.5 | 0.6 | 0.6 | 0.5 |
| Scheduler Patience | 5 | 5 | 5 | 5 |
| Pretrained Model | bert-base-uncased | bert-base-uncased | bert-base-uncased | bert-base-chinese |

### B.4 FEATURE EXTRACTION

For IEMOCAP dataset, in line with previous studies (Lian et al., 2023; Fu et al., 2024), we apply pre-trained DeBERTa (He et al., 2020) to encode word sequences into 1024-dimensional texture embeddings for each utterance, while MA-Net (Zhao et al., 2021) and wav2vec (Schneider et al., 2019) are used to extract visual and acoustic features, respectively.

For CMU-MOSI, CMU-MOSEI, and CH-SIMS dataset, our multimodal feature extraction process is consistent with previous studies (Li et al., 2023; Wang et al., 2023) by applying MMSA-FET Toolkit (Yu et al., 2020; 2021) to extract features.

- **Text Modality:** For English datasets, we utilize the BERT-base-uncased model to extract 768-dimensional hidden states. For CH-SIMS, we apply a BERT-based-Chinese model. These pre-trained language models capture rich contextual semantics, improving sentiment representation.

- **Visual Modality:** We extract facial action features using the OpenFace toolkit's Facet module, obtaining a 35-dimensional visual feature vector. These features capture facial expressions and microexpressions, which are essential for sentiment recognition.
- **Acoustic Modality:** We employ COVAREP to extract a 74-dimensional acoustic feature vector. These features include pitch, energy, and spectral properties, which are crucial for identifying speech-related emotional cues.

## B.5 Evaluation Metric for IEMOCAP

To evaluate our proposed method on the IEMOCAP dataset, we adopt the following evaluation metrics. Denote $C_0$ as the number of emotion classes in the dataset, and $\Gamma_j$ as the number of samples in class $j \in [1, C_0]$. Let $Acc_j$ and $F1_j$ represent the classification accuracy and the F1 score of class $j$, respectively.

Weighted average accuracy (WAcc) is a weighted mean accuracy over different emotion classes, with weights proportional to the number of utterances in a particular emotion class. It is defined as:

$$WAA = \frac{\sum_{j=1}^{C_0} \Gamma_j \cdot Acc_j}{\sum_{j=1}^{C_0} \Gamma_j} \tag{14}$$

Similarly, weighted average F1 Score (WAF1) is a weighted mean F1 score over different emotion categories, using weights proportional to the number of utterances in each emotion class:

$$WAF1 = \frac{\sum_{j=1}^{C_0} \Gamma_j \cdot F1_j}{\sum_{j=1}^{C_0} \Gamma_j} \tag{15}$$

The IEMOCAP dataset consists of discrete emotion categories. To ensure a fair comparison with existing methods, we evaluate emotion recognition performance using both weighted average accuracy (WAcc) and weighted average F1-score (WAF1).

## B.6 Statistical Estimation of Modality-Common Features

**Mean:** $\mu_{com}^{(m)} = \frac{1}{N} \sum_{n=1}^{N} \mathbf{f}_n^{(m)}$

**Covariance:** $\Sigma_{com}^{(m)} = \frac{1}{N} \sum_{n=1}^{N} (\mathbf{f}_n^{(m)} - \mu_{com}^{(m)})(\mathbf{f}_n^{(m)} - \mu_{com}^{(m)})^\top$

**Skewness:** $\Gamma_{com}^{(m)} = \frac{1}{N} \sum_{n=1}^{N} \left( \frac{\mathbf{f}_n^{(m)} - \mu_{com}^{(m)}}{\sqrt{\text{diag}(\Sigma_{com}^{(m)}) + \epsilon}} \right)^3$

## C Additional Experimental Analysis

### C.1 Extended Analysis for CMU-MOSI and CMU-MOSEI

Tables 5 and 6 present the extended comparison results on the CMU-MOSI and CMU-MOSEI datasets. Several insightful trends can be observed.

**Early Fusion and Transformer-based Methods.** Early multimodal fusion approaches such as MFM (Tsai et al., 2018) and MulT (Tsai et al., 2019) achieve competitive baseline results, demonstrating the effectiveness of early cross-modal attention mechanisms. However, these models exhibit limitations in capturing fine-grained interactions, particularly when modality-specific features interfere with global semantics. For example, MulT improves over MFM by explicitly modeling directional cross-attention, yet still suffers from modality imbalance and over-reliance on dominant modalities.

**Feature Decoupling and Disentanglement.** Subsequent models such as MISA (Hazarika et al., 2020), CENet (Wang et al., 2022), and Self-MM (Yu et al., 2021) introduce feature disentanglement or modality-invariant modeling to alleviate semantic interference. By separating modality-common and

modality-specific features, these approaches achieve steady gains in both correlation and classification metrics. For instance, Self-MM attains a relatively high Corr (0.764) and F1 score (83.04) on MOSI, while FDMER (Yang et al., 2022) further enhances disentangled representation learning through factorized modeling, pushing the F1 score to 83.22.

**Pre-trained Models and Advanced Alignment.** Recent advances such as AOBERT (Kim & Park, 2023), DMD (Li et al., 2023), and CGGM (Guo et al., 2025) leverage pre-trained language models and sophisticated alignment strategies. AOBERT benefits from the representational power of BERT for textual features, while CGGM integrates classifier-guided generative modeling to better capture distributional patterns. DMD, in particular, achieves strong results across both regression and classification tasks, with MOSI (MAE = 0.744, Acc-2 = 83.24, F1 = 83.55) and MOSEI (MAE = 0.561, Acc-2 = 84.17, F1 = 83.88), showing the advantage of graph-based knowledge distillation for balancing modality-specific and modality-shared information.

**Our DecAlign.** Most importantly, our proposed DecAlign consistently outperforms all baselines by a notable margin. On MOSI, DecAlign achieves the lowest MAE (0.735), the highest Corr (0.811), and significant improvements in both Acc-2 (85.75) and Acc-7 (45.07), as well as F1 score (85.82). Similarly, on MOSEI, DecAlign sets new benchmarks with MAE = 0.543, Corr = 0.768, Acc-2 = 86.48, Acc-7 = 55.02, and F1 = 86.07. These improvements of around 2–3 points in Acc-2 and F1 score compared to the strongest baselines (e.g., DMD and CGGM) highlight three key strengths: ❶ the ability of prototype-guided optimal transport to capture fine-grained heterogeneous interactions, ❷ the effectiveness of hierarchical alignment in balancing modality-common and modality-unique representations, and ❸ improved robustness against modality imbalance, enabling DecAlign to model subtle multimodal signals more faithfully.

| Models | MAE ($\downarrow$) | Corr ($\uparrow$) | Acc-2 ($\uparrow$) | Acc-7 ($\uparrow$) | F1 Score ($\uparrow$) |
|---|---|---|---|---|---|
| MFM (Tsai et al., 2018) | 0.951 | 0.662 | 78.18 | 36.21 | 78.10 |
| MulT (Tsai et al., 2019) | 0.846 | 0.725 | 81.70 | 40.05 | 81.66 |
| PMR (Fan et al., 2023) | 0.895 | 0.689 | 79.88 | 40.60 | 79.83 |
| CubeMLP (Sun et al., 2022) | 0.838 | 0.695 | 81.85 | 41.03 | 81.74 |
| MUTA-Net (Tang et al., 2023) | 0.767 | 0.736 | 82.12 | 40.88 | 82.07 |
| MISA (Hazarika et al., 2020) | 0.788 | 0.744 | 82.07 | 41.27 | 82.43 |
| CENet (Wang et al., 2022) | 0.745 | 0.749 | 82.40 | 41.32 | 82.56 |
| Self-MM (Yu et al., 2021) | 0.765 | 0.764 | 82.88 | 42.03 | 83.04 |
| FDMER (Yang et al., 2022) | 0.760 | 0.777 | 83.01 | 42.88 | 83.22 |
| AOBERT (Kim & Park, 2023) | 0.780 | 0.773 | 83.03 | 43.21 | 83.02 |
| DMD (Li et al., 2023) | 0.744 | 0.788 | 83.24 | 43.88 | 83.55 |
| ReconBoost (Hua et al., 2024) | 0.793 | 0.769 | 82.59 | 42.70 | 82.72 |
| CGGM (Guo et al., 2025) | 0.787 | 0.792 | 82.73 | 43.47 | 82.89 |
| DecAlign (Ours) | **0.735** | **0.811** | **85.75** | **45.07** | **85.82** |

Table 5: Performance Comparison on CMU-MOSI dataset. $\uparrow$ and $\downarrow$ indicate that higher or lower value is better. Best results are highlighted in **bold**, and suboptimal results are underlined. All reported results are averaged over **five** runs on the test set.

## C.2 EXTENDED ANALYSIS FOR CH-SIMS

Table 7 reports the results on the CH-SIMS dataset (Yu et al., 2020), which is particularly challenging due to its Chinese language modality and the greater diversity of sentiment expressions. Several key observations can be made.

**Early Fusion and Transformer-based Models.** Traditional fusion models such as MFM (Tsai et al., 2018) and MulT (Tsai et al., 2019) provide reasonable baselines, showing that cross-attention mechanisms are effective in modeling interactions across modalities. However, their performance is limited by over-reliance on direct fusion, which often struggles to capture the nuanced sentiment variations inherent in Chinese multimodal data.

**Disentanglement and Modality-specific Modeling.** Models such as MISA (Hazarika et al., 2020), CENet (Wang et al., 2022), and FDMER (Yang et al., 2022) achieve steady improvements by explicitly modeling modality-invariant and modality-specific features. This disentanglement reduces semantic interference and enables more precise sentiment prediction. For example, FDMER demonstrates

| Models | MAE (↓) | Corr (↑) | Acc-2 (↑) | Acc-7 (↑) | F1 Score (↑) |
|---|---|---|---|---|---|
| MFM (Tsai et al., 2018) | 0.681 | 0.555 | 78.93 | 45.93 | 76.45 |
| MulT (Tsai et al., 2019) | 0.673 | 0.677 | 80.85 | 48.37 | 80.86 |
| PMR (Fan et al., 2023) | 0.645 | 0.689 | 81.57 | 48.88 | 81.56 |
| CubeMLP (Sun et al., 2022) | 0.601 | 0.701 | 81.36 | 49.07 | 81.75 |
| MUTA-Net (Tang et al., 2023) | 0.617 | 0.717 | 81.76 | 49.88 | 82.01 |
| MISA (Hazarika et al., 2020) | 0.594 | 0.724 | 82.03 | 51.43 | 82.13 |
| CENet (Wang et al., 2022) | 0.588 | 0.738 | 82.13 | 52.31 | 82.35 |
| Self-MM (Yu et al., 2021) | 0.576 | 0.732 | 82.43 | 52.68 | 82.47 |
| FDMER (Yang et al., 2022) | 0.571 | 0.743 | 83.88 | 53.21 | 83.35 |
| AOBERT (Kim & Park, 2023) | 0.588 | 0.738 | 83.90 | 52.47 | 83.14 |
| DMD (Li et al., 2023) | 0.561 | 0.758 | 84.17 | 54.18 | 83.88 |
| ReconBoost (Hua et al., 2024) | 0.599 | 0.733 | 82.98 | 52.68 | 83.14 |
| CGGM (Guo et al., 2025) | 0.584 | 0.760 | 83.72 | 52.88 | 83.94 |
| DecAlign (Ours) | **0.543** | **0.768** | **86.48** | **55.02** | **86.07** |

Table 6: Performance Comparison on CMU-MOSEI dataset. ↑ and ↓ indicate that higher or lower value is better. Best results are highlighted in **bold**, and suboptimal results are underlined. All reported results are averaged over **five** runs on the test set.

robust performance with reduced MAE and improved F1, confirming the benefits of factorized disentanglement for sentiment modeling.

**Advanced Alignment and Classifier-guided Fusion.** Recent methods such as DMD (Li et al., 2023), MCIS (Zhou et al., 2025), and CGGM (Guo et al., 2025) deliver stronger baselines by leveraging more sophisticated alignment strategies. DMD and MCIS achieve MAE around 0.421–0.429 with F1 scores close to 80, showing balanced regression and classification performance. Notably, CGGM achieves the best results among existing baselines (MAE = 0.417, F1 = 80.12, Acc-3 = 80.17, Corr = 0.638), highlighting the effectiveness of classifier-guided generative modeling in capturing sentiment-related cues in Chinese multimodal contexts.

**Our DecAlign.** Despite these advances, DecAlign consistently achieves superior performance across all metrics, with MAE = 0.403, F1 = 81.85, Acc-3 = 88.24, and Corr = 0.657. These gains are substantial: in particular, the +8 point improvement in Acc-3 compared to the strongest baseline (CGGM) underscores DecAlign's ability to achieve fine-grained categorical sentiment classification. The consistent reduction in MAE and higher correlation also indicate that DecAlign better captures continuous sentiment intensity.

**Key Insights.** The results validate three important properties of DecAlign: ❶ *Cross-lingual robustness*: by successfully handling Chinese sentiment data, DecAlign demonstrates strong generalization beyond English benchmarks. ❷ *Hierarchical alignment effectiveness*: prototype-guided optimal transport and latent semantic consistency work synergistically to reduce modality interference and enhance categorical prediction. ❸ *Balanced regression and classification*: DecAlign achieves state-of-the-art results in both continuous and discrete tasks, reflecting its ability to model both sentiment strength and polarity.

## C.3 EXTENDED ANALYSIS FOR ABLATION STUDIES

We provide an in-depth analysis of the ablation results in Table 2 and the visualizations in Figure 4. We examine both the impact of *key components*—Multimodal Feature Decoupling (MFD), Heterogeneity Alignment (Hete), and Homogeneity Alignment (Homo)—and the contribution of *specific strategies* within the alignment modules, namely Prototype-guided Optimal Transport (Proto-OT), Contrastive Training (CT), Semantic Consistency (Sem), and Maximum Mean Discrepancy regularization (MMD).

**A. Key components: MFD, Hete, Homo.**

❶ *Decoupling is a prerequisite, alignment is the catalyst.* When only MFD is retained (i.e., both Hete and Homo are removed), the model degrades substantially (MOSI: MAE 0.784, F1 81.92; MOSEI: MAE 0.632, F1 82.22). This indicates that decoupling alone, while preventing raw feature

| Models | MAE ($\downarrow$) | F1 Score ($\uparrow$) | Acc-3 ($\uparrow$) | Corr ($\uparrow$) |
|---|---|---|---|---|
| MFM (Tsai et al., 2018) | 0.471 | 75.28 | 75.32 | 0.516 |
| MulT (Tsai et al., 2019) | 0.455 | 76.96 | 77.02 | 0.544 |
| PMR (Fan et al., 2023) | 0.445 | 76.55 | 76.63 | 0.523 |
| CubeMLP (Sun et al., 2022) | 0.459 | 77.85 | 77.94 | 0.562 |
| MUTA-Net (Tang et al., 2023) | 0.443 | 77.21 | 77.44 | 0.573 |
| MISA (Hazarika et al., 2020) | 0.437 | 78.43 | 78.56 | 0.581 |
| CENet (Wang et al., 2022) | 0.454 | 78.03 | 78.15 | 0.589 |
| Self-MM (Yu et al., 2021) | 0.432 | 77.97 | 78.03 | 0.593 |
| FDMER (Yang et al., 2022) | 0.424 | 78.74 | 78.82 | 0.599 |
| AOBERT (Kim & Park, 2023) | 0.430 | 78.55 | 78.65 | 0.578 |
| DMD (Li et al., 2023) | 0.421 | 79.88 | 78.98 | 0.612 |
| MCIS (Yang et al., 2024) | 0.429 | 79.58 | 79.64 | 0.629 |
| CGGM (Guo et al., 2025) | 0.417 | 80.12 | 80.17 | 0.638 |
| DecAlign (Ours) | **0.403** | **81.85** | **88.24** | **0.657** |

Table 7: Performance comparison on the CH-SIMS dataset. $\uparrow$ indicates that higher values are better. Best results are highlighted in **bold**, and runner-up results are underlined.

entanglement, cannot suppress modality-unique interference nor enforce cross-modal semantic coherence. Enabling both alignments in the full model reduces error and boosts classification notably (MOSI: MAE **0.735**, F1 **85.82**; MOSEI: MAE **0.543**, F1 **86.07**), yielding absolute gains over the MFD-only setting of $\Delta$MAE=0.049/0.089 and $\Delta$F1=3.90/3.85 on MOSI/MOSEI, respectively. This establishes that *decoupling prepares the space, while alignment shapes it.*

❷ *Heterogeneity alignment drives regression error down; homogeneity alignment stabilizes classification.* With **Hete**-only (no Homo), the model achieves lower MAE and higher F1 than Homo-only: MOSI (MAE 0.747, F1 84.46) vs Homo-only (MAE 0.754, F1 84.03); MOSEI (MAE 0.562, F1 84.74) vs Homo-only (MAE 0.588, F1 84.37). This shows that addressing *distributional mismatch* via Hete (Proto-OT + transformer refinement) has a more direct impact on minimizing continuous error (MAE), while Homo alignment (Sem + MMD) contributes more to *global semantic smoothing and decision stability*. In short, Hete is the primary lever for regression fidelity; Homo ensures calibrated and consistent boundaries.

❸ *Hierarchical complementarity is essential, especially for subtle sentiments.* Figure 4(a)–(d) reveals that removing either Hete or Homo produces uneven drops across categories, with pronounced degradation at {-1, 0, +1} sentiment levels. This pattern suggests that the two alignments address complementary failure modes: Hete mitigates local structure mismatches (reducing noisy shifts around decision margins), while Homo enforces global semantic agreement (preventing over-fragmented boundaries). Their combination recovers both *fine-grained discrimination* and *global robustness*.

❹ *From modality gap to semantic co-location.* The t-SNE visualizations in Figure 4(e)–(h) corroborate the above: without alignment, paired language–vision features are distant with erratic pairwise directions; with Homo-only, clusters get closer but remain fragmented; with Hete-only, pairwise distances shrink further but residual anisotropy persists. Only the full model yields *tight, co-located* clusters, reflecting both local and global alignment. This explains the joint improvements in MAE and F1.

**B. Strategy-level analysis: Proto-OT, CT, Sem, MMD.**

❶ *Prototype-guided Optimal Transport (Proto-OT) is the backbone for error reduction.* Removing Proto-OT leads to marked MAE increases (MOSI: 0.748; MOSEI: 0.624 vs full 0.735/0.543), and consistent F1 drops (MOSI: 84.17, MOSEI: 85.03). This shows that *distribution-aware, prototype-level* alignment is indispensable for resolving cross-modal heterogeneity, especially in regression where small misalignments accumulate into larger intensity errors.

❷ *Contrastive Training (CT) is crucial for discriminability and margin preservation.* Without CT, inter-class separation weakens (MOSI F1: 84.36; MOSEI F1: 85.21), and MAE also deteriorates

(MOSI: 0.743, MOSEI: 0.619). CT establishes *class-aware* anchors in the aligned space, preventing representation collapse and ensuring sharper decision boundaries—an effect that directly benefits Acc/F1 while indirectly stabilizing MAE by discouraging ambiguous embeddings near thresholds.

❸ *Semantic Consistency (Sem) aligns moments beyond the mean—vital for stable fusion.* Replacing Sem with only MMD (i.e., removing the explicit latent-moment matching) yields broader, more variable clusters and a noticeable performance drop (MOSI F1 84.73 vs full 85.82; MOSEI F1 85.33 vs full 86.07). Sem's explicit constraints on mean, covariance, and higher-order structure improve *global shape agreement* across modalities, thereby regularizing fusion and mitigating systematic biases.

❹ *MMD supplies non-parametric distributional regularization that smooths the latent geometry.* Omitting MMD also harms performance (MOSI: MAE 0.741, F1 84.61; MOSEI: MAE 0.564, F1 85.26). While its marginal effect may appear smaller than Proto-OT or CT, MMD complements Sem by *non-parametrically* matching distributions in RKHS, capturing high-order statistics and preventing overfitting to specific batch-level alignments.

❺ *Orthogonal roles, additive gains.* Comparing the four single-removal cases with the full model (MOSI: 0.735/85.82; MOSEI: 0.543/86.07), removing Proto-OT or CT most strongly hurts MAE and F1 (structure and margin), whereas removing Sem or MMD still impairs performance (global coherence and smoothness) but less severely. This ordering suggests that *structural alignment (Proto-OT)* and *discriminative supervision (CT)* form the core, while *semantic moment matching (Sem)* and *non-parametric regularization (MMD)* supply the necessary global consistency to fully realize the gains.

## C. Cross-metric and cross-dataset insights.

❶ *MAE vs F1: different facets of the same alignment.* Hete (via Proto-OT) predominantly reduces MAE, reflecting improved geometric co-registration of modality-unique structures; Homo (Sem+MMD) mainly consolidates F1 by smoothing the cross-modal manifold and calibrating decision surfaces. Their synergy explains the concurrent improvements in both regression and classification.

❷ *Dataset scale and heterogeneity matter.* On the larger and more diverse MOSEI, removing CT or Proto-OT incurs greater MAE penalties (+0.076/+0.081) than on MOSI (+0.008/+0.013), indicating that *class-aware structure* and *prototype-level transport* are especially critical when the data distribution is broader and more multimodal.

❸ *Category sensitivity and boundary sharpening.* Figure 4(a)–(d) shows that neutral and near-neutral bins suffer most when either alignment is removed. This suggests that hierarchical alignment is particularly effective at *sharpening ambiguous boundaries*, where cross-modal cues are subtle and easily swamped by dominant modality noise.

❹ *From local structure to global semantics.* The combined evidence from Table 2 and Figure 4 indicates a two-stage mechanism: Hete reduces local structural discrepancies (prototype geometry, density mismatch), while Homo imposes global semantic consistency (moment alignment, RKHS discrepancy). The full model closes the *local-to-global* loop, yielding robust improvements across all metrics.

## D. Takeaways for designing multimodal aligners.

❶ *Always decouple before you align.* MFD isolates modality-unique and -common factors, ensuring that subsequent alignment targets the right subspaces instead of wrestling with entangled representations.

❷ *Prioritize structure-aware alignment.* Prototype-level transport should be a first-class component: it is consistently the strongest driver of MAE reduction and a key stabilizer of downstream classification.

❸ *Do not trade off discriminability for alignment.* Contrastive supervision is necessary to maintain class margins during alignment; it prevents over-alignment that collapses inter-class structure.

❹ *Match distributions twice: parametrically and non-parametrically.* Moment-based Sem and kernel-based MMD play distinct roles and are most effective in tandem, aligning both *shape* and *support* of latent distributions.

Overall, the ablation evidence supports DecAlign's hierarchical design: structural alignment of heterogeneous factors (Proto-OT + transformer refinement) combined with semantic and distributional alignment of homogeneous factors (Sem + MMD) is *jointly necessary* to achieve the consistent, cross-metric improvements observed across benchmarks.

## C.4 Extended Analysis for Modality Gap

Figure 4 (e)–(h) visualizes the modality gap between language and vision features under different ablation settings. Each subplot presents paired features from the two modalities after projection into a 2D space using t-SNE, where lines connect corresponding language–vision pairs of the same input instance. Several key insights can be drawn.

**Lack of Alignment.** In subfigure (e), where both heterogeneity and homogeneity alignment modules are removed, the projected features from language and vision are widely separated with irregular pairwise connections. The high dispersion and inconsistent alignment directions indicate a severe modality gap, driven by the absence of constraints on either modality-unique discrepancies or cross-modal commonality. This validates that simple multimodal feature projection is insufficient for achieving semantic consistency.

**Effect of Homogeneity-only Alignment.** Subfigure (f), which removes the heterogeneity alignment but preserves homogeneity alignment, shows partial improvements: paired features are closer, and cluster overlap increases. However, disjoint sub-clusters remain visible, suggesting that enforcing semantic consistency via latent distribution matching reduces global misalignment but fails to resolve modality-specific variations. This highlights that semantic alignment alone cannot fully mitigate distributional heterogeneity.

**Effect of Heterogeneity-only Alignment.** In subfigure (g), where homogeneity alignment is removed but heterogeneity alignment is retained, the features are more concentrated and inter-modal distances shrink further. Prototype-guided optimal transport effectively aligns modality-unique structures by reducing distributional mismatch. Nevertheless, residual vertical dispersion across clusters reveals that without semantic alignment, global consistency is not guaranteed, leaving subtle but systematic modality biases uncorrected.

**Full DecAlign with Hierarchical Alignment.** Finally, subfigure (h) illustrates the full DecAlign framework with both alignment strategies enabled. Here, paired features are tightly clustered and nearly co-located, with minimal alignment distances and consistent cluster structures across modalities. This demonstrates the complementary effects of heterogeneity and homogeneity alignment: the former resolves modality-specific discrepancies at the distributional level, while the latter enforces semantic consistency in the latent space. Their combination closes the modality gap both locally and globally, leading to highly consistent cross-modal representations.

**Key Insights.** These visual analyses validate that *hierarchical alignment is crucial for robust multimodal integration*. Removing either module leads to partial but incomplete alignment, while the joint application of both substantially minimizes the modality gap. This explains why DecAlign achieves significant gains across benchmarks: it harmonizes modality-unique and modality-common representations simultaneously, ensuring that cross-modal signals are both semantically consistent and structurally coherent.

