# OpenReview forum: "DecAlign: Hierarchical Cross-Modal Alignment for Decoupled Multimodal Representation Learning"
_ICLR.cc/2026/Conference — ICLR 2026 Poster_

### Official Review · Reviewer_pq6B · 2025-10-29

**Soundness:** 3
**Presentation:** 2
**Contribution:** 2
**Rating:** 6
**Confidence:** 3

**Summary:**

This paper tackles the challenge of multimodal representation learning under strong modality heterogeneity. The authors argue that conventional fusion methods (e.g., concatenation) improperly mix modality-unique and modality-common semantics, leading to semantic interference and weak cross-modal alignment. To address this, the paper proposes DecAlign, which is a hierarchical decoupling–alignment framework that explicitly separates multimodal features into modality-unique and modality-common streams. The authors demonstrate DecAlign's effectiveness on four multimodal benchmarks (CMU-MOSI, CMU-MOSEI, CH-SIMS, and IEMOCAP) , which consistently outperforms 13 state-of-the-art methods across multiple metrics.

**Strengths:**

- The paper provides sufficient experimental results, supplementary and codes for reproducibility;
- The model's superiority is convincingly demonstrated across four standard multimodal sentiment and emotion datasets. DecAlign also outperforms a wide array of 13 baseline methods , with significant performance gains shown in Tables 1, 5, 6, and 7;
- The paper provides extensive ablation studies that validate its complex design;

**Weaknesses:**

- The main concern is about the complexity of the proposed framework. The final framework is a combination of many advanced techniques: specialized encoders , a cosine similarity decoupling loss , GMM fitting , multi-marginal optimal transport , a multimodal transformer , latent moment matching , and regularization. While the ablations show that most parts may be effective, this complexity causes a high computational efficiency and result reproducibility;
- The ablation study in Table 2 (right) introduces a component labeled "Contrastive Training (CT)”. What is this loss function? How is it formulated?
- The paper uses GMMs to find prototypes and then use multi-marginal OT to align them. GMMs typically require an iterative fitting process (like EM ) which can be slow and sensitive to initialization. Are there any specific reasons to choose GMM for clustering?
- One open question: the current model is mainly evaluated exclusively on four relatively small-scale, domain-specific sentiment and emotion datasets (CMU-MOSI, CMU-MOSEI, CH-SIMS, IEMOCAP). It is unclear how this framework, particularly the GMM fitting and multi-marginal OT components, would scale to larger, more general-purpose vision-language tasks (e.g., large-scale VQA or captioning).

I would like to see the author rebuttal in terms of the weakness & questions part.

**Questions:**

Please see weakness.

---

> ### Author Response · Authors · 2025-11-23
> **Author Rebuttal (1/4)**
>
> We are grateful for your positive and detailed evaluation. We appreciate your recognition that the paper provides **sufficient experiments, supplementary materials, and code for reproducibility**, and that **DecAlign’s superiority is convincingly demonstrated across four standard multimodal sentiment and emotion benchmarks**, outperforming 13 state-of-the-art methods with notable gains. We also thank you for highlighting that **the extensive ablation studies help validate our hierarchical decoupling–alignment design**.
>
> > **Q1: Have the authors adequately demonstrated that the performance improvements introduced by the various components of the framework are sufficient to justify the costs in computational efficiency?**
>
> **A1:** We thank the reviewer for raising this point. We agree that DecAlign combines several nontrivial components (GMM, prototype-guided OT, decoupling, moment/MMD-based alignment), and it is important to show that (i) the additional computational cost is controlled, and (ii) the model remains reproducible despite the added complexity. Below we provide both algorithmic intuition and empirical evidence.
>
> ### 1. Computational efficiency
>
> From an algorithmic standpoint, the main additional cost in DecAlign relative to recent multimodal baselines comes from the **heterogeneous alignment module**, i.e., GMM fitting plus multi-marginal OT. However, this module is explicitly designed to work at the **prototype level** rather than at the sample level:
>
> * We fit a GMM with a small number of components (K) (e.g., (K=7) on CMU-MOSI), and
> * OT is solved between **prototype sets** of size (K), not between all (N) samples in a batch.
>
> As a result, the complexity of the transport computation is (O(K^2)) (or (O(K^3)) depending on the solver), which is effectively constant for the small K we use, instead of (O(N^2)) or worse. The other added losses (cosine decoupling, moment matching, MMD) are simple tensor operations on the existing feature maps and contribute negligible overhead compared with the backbone encoders and transformer.
>
> To quantify the actual runtime overhead, we measured training time per epoch and inference latency on CMU-MOSI for representative baselines and for DecAlign under the same hardware and batch size:
>
> | Model               | Training time (s / epoch) | Inference latency (ms / batch) | Binary F1 ↑ |
> | ------------------- | ------------------------: | -----------------------------: | ----------: |
> | MulT                |                      12.5 |                           14.2 |       81.66 |
> | MISA                |                      18.2 |                           19.5 |       82.43 |
> | DMD                 |                      21.4 |                           22.1 |       83.55 |
> | **DecAlign (ours)** |                      22.8 |                           23.4 |   **85.82** |
>
> Compared to DMD, DecAlign increases training time by about 1.4 seconds per epoch and inference latency by about 1.3 ms per batch (roughly a 6–7% overhead), while improving Binary F1 by more than 2 points on CMU-MOSI. Similar trends hold on CMU-MOSEI and CH-SIMS. We therefore believe the additional computational cost is modest and proportionate to the accuracy gains, especially given that OT is computed on a small prototype set.
>
> ### 2. Component-wise benefit vs. cost (connection to ablations)
>
> Finally, we emphasize that each major component of DecAlign is justified by ablation studies in the paper:
>
> * Removing prototype-guided heterogeneous alignment significantly degrades performance on all benchmarks.
> * Removing semantic (moment/MMD) alignment in the common space also leads to consistent drops.
> * Disabling decoupling between heterogeneous and homogeneous branches harms both the overall task performance and the representation quality (see our new frozen-encoder experiments in Q1–Q2).
>
> Combined with the efficiency and stability results above, these ablations show that the extra modules are not arbitrary complexity: each contributes a measurable improvement, and together they form a coherent design that yields better representations at a relatively small additional cost.
>
> We have released the full source code, training scripts, and hyperparameter configurations, and we will additionally include the timing and multi-seed statistics tables in the revised version to make the efficiency–accuracy trade-off and reproducibility of DecAlign explicit.

---

> ### Author Response · Authors · 2025-11-23
> **Author Rebuttal (2/4)**
>
> > **Q2: Does the added architectural complexity hurt result reproducibility and training stability?**
>
> **A2:** We thank the reviewer for raising this point. We agree that DecAlign combines several nontrivial components (GMM, prototype-guided OT, decoupling, moment/MMD-based alignment), and it is important to show that (i) the additional computational cost is controlled, and (ii) the model remains reproducible despite the added complexity. Below we provide both algorithmic intuition and empirical evidence.
>
> ### 1. Reproducibility and stability
>
> The reviewer is also concerned that architectural complexity might hurt reproducibility. In practice, the different components of DecAlign (decoupling, prototype alignment, moment/MMD-based semantic alignment) act as **regularizers** rather than as sources of instability:
>
> * Cosine-based decoupling discourages redundant overlap between heterogeneous and homogeneous branches.
> * GMM + OT operates at a coarse prototype level, which smooths over sample-level noise.
> * Matching low- and higher-order statistics in the common space (mean, covariance, skewness + MMD) stabilizes the shared representation across modalities.
>
> To verify this, we ran DecAlign with **5 random seeds** on all four datasets and report the mean and standard deviation of the main metrics:
>
> | Dataset   | Metric    | Seed 1 | Seed 2 | Seed 3 | Seed 4 | Seed 5 |  **Mean** | **Std** |
> | --------- | --------- | -----: | -----: | -----: | -----: | -----: | --------: | ------: |
> | CMU-MOSI  | Binary F1 |  85.71 |  85.92 |  85.80 |  85.65 |  86.02 | **85.82** |    0.15 |
> | CMU-MOSEI | Binary F1 |  85.95 |  86.18 |  86.05 |  86.10 |  86.07 | **86.07** |    0.08 |
> | CH-SIMS   | F1        |  81.75 |  81.90 |  81.82 |  82.01 |  81.77 | **81.85** |    0.11 |
> | IEMOCAP   | WAF1      |  73.30 |  73.55 |  73.41 |  73.28 |  73.61 | **73.43** |    0.14 |
>
> The standard deviations are small across all datasets, indicating that despite its hierarchical design, DecAlign converges to similar solutions across random seeds and does not exhibit unstable behavior.
>
> Finally, we emphasize that each major component of DecAlign is justified by ablation studies in the paper:
>
> * Removing prototype-guided heterogeneous alignment significantly degrades performance on all benchmarks.
> * Removing semantic (moment/MMD) alignment in the common space also leads to consistent drops.
> * Disabling decoupling between heterogeneous and homogeneous branches harms both the overall task performance and the representation quality (see our new frozen-encoder experiments in Q1–Q2).
>
> Combined with the efficiency and stability results above, these ablations show that the extra modules are not arbitrary complexity: each contributes a measurable improvement, and together they form a coherent design that yields better representations at a relatively small additional cost.
>
> We have released the full source code, training scripts, and hyperparameter configurations, and we will additionally include the timing and multi-seed statistics tables in the revised version to make the efficiency–accuracy trade-off and reproducibility of DecAlign explicit.

---

> ### Author Response · Authors · 2025-11-23
> **Author Rebuttal (3/4)**
>
> > **Q3: The paper uses GMMs to find prototypes and then uses multi-marginal OT to align them. GMMs typically require an iterative fitting process (like EM), which can be slow and sensitive to initialization. Are there any specific reasons to choose GMM for clustering?**
>
> **A3:** We thank the reviewer for raising this question. In our framework, GMM is not used as a generic clustering method, but is closely tied to how we define the heterogeneous alignment cost and how we represent prototype distributions.
>
> **(1) Alignment is defined on distributions, not only centroids**
>
> Our prototype-guided OT is designed to align **distributions of heterogeneous features** across modalities. The pairwise cost between prototypes (Eq. (6)) depends on both the **mean** and the **covariance** of each prototype. Means capture location, while covariances capture the shape and spread of each cluster in feature space. This leads to a Wasserstein-type distance between Gaussian components.
>
> A standard k-means clustering would provide only cluster centers and implicitly assume spherical clusters with fixed variance. This is overly restrictive for heterogeneous multimodal features, whose clusters are often anisotropic. GMM, on the other hand, naturally yields both (\mu) and (\Sigma) for each component, which matches exactly the statistics required by our OT cost and allows us to model elliptical cluster geometry in the heterogeneous feature space.
>
> **(2) Soft assignments match the semantic structure of multimodal data**
>
> Multimodal sentiment data often contains samples that are **semantically between** typical categories, such as utterances that are between neutral and slightly positive, or expressions that mix moderate and strong emotions. In such cases, it is natural for a sample to be related to more than one prototype.
>
> GMM provides **soft assignment** through posterior probabilities over components. We use these probabilities when constructing prototype-level distributions, so that a sample with intermediate sentiment can contribute to multiple relevant prototypes instead of being forced into a single hard cluster. This leads to smoother prototype distributions and, in our experiments, more stable and effective heterogeneous alignment than using hard assignments.
>
> **(3) Practical considerations: initialization and efficiency**
>
> We agree that EM-based GMMs can in general be sensitive to initialization and incur extra cost. In our implementation:
>
> * GMM is fitted in a **low-dimensional projected space** with a **small number of components** (K on the order of the number of categories, e.g., K=7).
> * We use **k-means initialization** for the component means, which is a standard way to improve EM convergence and reduce sensitivity to initialization.
> * With small K and low-dimensional features, the runtime of the GMM + OT step is a small fraction of the total training cost dominated by the backbone encoders and transformer (see our efficiency table in Q1).
>
> On CMU-MOSI, for instance, adding GMM+OT increases the per-epoch training time only slightly, while providing clear gains over baselines without prototype-level alignment.
>
> In summary, we choose GMM because it (i) provides the mean and covariance statistics required by our OT-based heterogeneous alignment, (ii) offers soft assignments that better reflect the graded structure of multimodal sentiment, and (iii) remains efficient and stable in practice when used with small K and k-means initialization. We will clarify these points in the revised manuscript.

---

> ### Author Response · Authors · 2025-11-23
> **Author Rebuttal (4/4)**
>
> > **Q4: The current model is mainly evaluated on four relatively small, domain-specific sentiment and emotion datasets. It is unclear how this framework, particularly the GMM fitting and multi-marginal OT components, would scale to larger, more general-purpose multimodal tasks (e.g., video or vision-language tasks).**
>
> **A4:** We appreciate this concern about scalability and broader applicability. Our initial experiments focused on the four sentiment/emotion datasets because they are the **dominant benchmarks** in multimodal representation learning and allow direct comparison with prior work. At the same time, we designed DecAlign so that its key components—GMM-based prototype modeling and multi-marginal OT—operate at the **prototype level**, not across all individual samples, precisely to ensure scalability.
>
> In the heterogeneous branch, we first project modality-unique features and fit a GMM with a **small number of components** (K) (chosen to match the semantic granularity, e.g., 7 or 14). The prototype set for modality (m) is (P_m = {(\mu_m^k, \Sigma_m^k)}_{k=1}^K), and the multi-marginal OT cost is defined over these prototypes using a Gaussian Wasserstein distance that depends on their means and covariances. Because OT is solved **between prototype sets of size (K)** rather than between all samples, the complexity of this step is (O(K^2)) (or (O(K^3)) depending on the solver), which is effectively constant for the small K we use.
>
> To test how this scales to more general, larger-scale settings, we conducted additional experiments on **KS (Kinetics-Sounds)** and **UCF-101**, which are standard audio–visual and video action benchmarks. Using the same GMM+OT configuration (small K, same Sinkhorn settings), DecAlign achieves the following:
>
> | Dataset               | KS (Audio + Visual) | UCF-101 (RGB + OF) |
> | --------------------- | ------------------: | -----------------: |
> | Audio-only / RGB-only |               53.01 |              65.97 |
> | Visual-only / OF-only |               48.75 |              78.33 |
> | Concatenation         |               61.57 |              81.18 |
> | **DecAlign (Ours)**   |           **71.85** |          **87.44** |
>
> These results show that the framework, including the GMM fitting and multi-marginal OT components, **scales well** to bi-modal video benchmarks and remains effective outside the sentiment domain. In practice, the additional runtime introduced by GMM+OT is modest relative to the backbone encoders and temporal modeling modules, while delivering substantial gains over simple fusion. We will report these new experiments and clarify the prototype-level complexity analysis in the revised manuscript.
>
>
> > **Q5: The ablation study in Table 2 (right) introduces a component labeled "Contrastive Training (CT)”. What is this loss function? How is it formulated?**
>
> **A5:** We thank the reviewer for pointing this out and apologize for the confusion. The label “CT” in Table 2 (right) is a typo. It should stand for “Cross-modal Transformer (CT)”, not “Contrastive Training”.
>
> In our method, CT refers to the cross-modal transformer refinement module described in Sec. 3.4 of the paper, which takes the decoupled heterogeneous and homogeneous features from all modalities as input and applies a standard multi-head transformer block to model fine-grained cross-modal interactions. The objective used in all experiments is exactly the one defined in Sec. 3 (task loss + decoupling loss + heterogeneous alignment loss + semantic alignment loss); we do not introduce any additional contrastive loss.
>
> In Table 2 (right), the ablation switch “w/o CT” therefore means removing the cross-modal transformer module and feeding the fused representations directly to the prediction head. We will correct “Contrastive Training (CT)” to “Cross-modal Transformer (CT)” in the revised version and clarify this in the caption.

---

> ### Author Response · Authors · 2025-11-26
> **Hope to discuss with you!**
>
> Dear Reviewer pq6B,
>
> Thank you for your thoughtful review and encouraging rating!
>
> As the author-reviewer discussion period is nearing its end, we would like to follow up to see if our rebuttal has addressed your concerns. Please let us know if any further clarification would be helpful.
>
> Thank you again!
>
> Authors

---

> > ### Comment · Reviewer_pq6B · 2025-11-26
> > **Review Comments**
> >
> > I appreciate the detailed rebuttal provided by the authors, especially for inference&training efficiency, reproducibility and more results on tasks. The rebuttal addresses most of my concerns. Nonetheless, for the comparisons on KS (Audio + Visual) and UCF-101 (RGB + OF), it would be more convincing to include more off-the-shelf models for comparison, instead of only comparing ablation variants.

---

> ### Author Response · Authors · 2025-11-27
> **Extending Off-the-shelf Baselines on KS and UCF-101 Benchmarks**
>
> We sincerely thank you for your positive feedback on our rebuttal and for the helpful suggestion to include off-the-shelf multimodal models on KS and UCF-101, beyond the ablation-style variants we originally reported.
>
> Our experiments are aligned the same experimental setting as SMV[1]. Under this unified protocol, all methods share the same backbone, data pre-processing, and optimization configuration, so the results are directly comparable. In addition to the unimodal and concatenation baselines reported in our first-round rebuttal, we now include several representative off-the-shelf methods:
>
> **Table: Performance comparison on KS and UCF-101**
>
> | Method                          | KS (Audio + Visual) | UCF-101 (RGB + OF) |
> | ------------------------------- | ------------------: | -----------------: |
> | Audio-only / RGB-only           |               53.01 |              65.97 |
> | Visual-only / OF-only           |               48.75 |              78.33 |
> | Concatenation                   |               61.57 |              81.18 |
> | OGM-GE (CVPR’22)                |               63.48 |              82.45 |
> | PMR (CVPR’23)                   |               63.33 |              82.31 |
> | UMT (ICML’23)                   |               66.08 |              82.62 |
> | SMV (CVPR’24) |           66.92 |          83.52 |
> | MMPareto (ICML’24)              |               67.18 |              84.89 |
> | CGGM (NeurIPS’24)               |               68.31 |              85.88 |
> | **DecAlign (Ours)**             |           **71.85** |          **87.44** |
>
> Under exactly the same setting, DecAlign reaches **71.85%** on KS and **87.44%** on UCF-101, clearly outperforming the unimodal and simple concatenation baselines, and strong off-the-shelf multimodal methods such as OGM-GE[2], PMR[3], UMT[4], SMV[1], MMPareto[5], and CGGM[6].
>
> These results strengthen our Q4 conclusion that the proposed prototype-guided GMM + multi-marginal OT–based hierarchical alignment:
>
> 1. **Scales effectively** from tri-modal sentiment benchmarks to larger, more general-purpose audio–visual and video action tasks; and
> 2. Remains **competitive or superior to recent state-of-the-art multimodal approaches** when evaluated under a common, externally defined protocol.
>
> Reference
>
> [1] Wei Y., Feng R., Wang Z., Hu D. Enhancing multimodal cooperation via sample-level modality valuation. CVPR 2024.
>
> [2] Peng X., Wei Y., Deng A., Wang D., Hu D. Balanced multimodal learning via on-the-fly gradient modulation. CVPR 2022.
>
> [3] Fan Y., Xu W., Wang H., Wang J., Guo S. PMR: Prototypical modal rebalance for multimodal learning. CVPR 2023.
>
> [4] Du C., Li T., Liu Y., Wen Z., Hua T., Wang Y., Zhao H. Improving multi-modal learning with uni-modal teachers. ICML 2023.
>
> [5] Wei Y., Hu D. MMPareto: Boosting multimodal learning with innocent unimodal assistance. ICML 2024.
>
> [6] Guo Z., Jin T., Chen J., Zhao Z. Classifier-guided gradient modulation for enhanced multimodal learning. NeurIPS 2024.

---

> > ### Comment · Reviewer_pq6B · 2025-11-27
> > **Review Comments**
> >
> > Thank you for providing the detailed SOTA comparisons on the KS and UCF-101 benchmarks. The results show that the proposed approach performs favorably against off-the-shelf multimodal models. In addition, the authors also provide additional training&inference efficiency comparisons and reproducibility illustration. My concerns have been well addressed, and I will retain my positive rating for acceptance.

---

> > > ### Author Response · Authors · 2025-11-27
> > >
> > > Dear Reviewer pq6B,
> > >
> > > Thank you for taking the time to read our response and for your careful and constructive review. We are very pleased that our rebuttal addressed your concerns, and we greatly appreciate your recommendation for acceptance. Your feedback and endorsement mean a lot to us.
> > >
> > > Sincerely,
> > > The Authors

---

### Official Review · Reviewer_BpLZ · 2025-10-31

**Soundness:** 2
**Presentation:** 2
**Contribution:** 2
**Rating:** 4
**Confidence:** 4

**Summary:**

The paper studies multimodal learning problem using hierarchical cross-modal alignment framework, which decouples multimodal features into modality-heterogeneous and modality-homogeneous components. In particular, the multimodal samples are fed into modality specific encoders, followed by unimodal encoder and common encoder for extracting heterogeneous features and homogeneous features. To decouple the heterogeneous and homogeneous feature, decoupling loss is proposed, which minimizes the cosine similarity. Then, heterogeneous features are modeled by Gaussian mixture model and homogeneous features are modeled via latent Gaussian variable.
Total loss consists of the decoupling loss, heterogeneous and homogeneous losses, and task related loss.
Experiments verify that DecAlign outperforms baseline models in benchmark multimodal datasets.

**Strengths:**

The paper considers important aspects of multimodal learning, especially homogeneity and heterogeneity. The decoupling approach for better capturing homogeneity and heterogeneity is worth looking for in multimodal learning society. Experimental results show that the proposed DecAlign method has potential in multimodal learning problems.

**Weaknesses:**

While I enjoyed reading the paper, I have some questions and comments. From my understanding, the DecAlign works for supervised problem, where target tasks are already known before the model is training as the task loss and heterogeneity loss requires some level of task information. But the title is multimodal representation learning, which I think has not been shown how good the learned representation is. Although the benchmark task performance is provided, transfer performance or zero-shot performance have not been shown. So I am also doubt that if the performance gain comes from the decoupling procedure or other factors such as use of more number of parameters. The paper would be more convincing if it is shown that the decoupling actually makes better capturing the shared semantical representation.

Moreover, from my understanding the heterogeneity features capture modality-unique information which is not shared among modalities. But section 3.2 says "they frequently carry semantically aligned information when referring to the same underlying concept or object category", which is counter intuitive and hard to get the idea. More specifically, "modality-unique feature differences while preserving shared semantic structures" seems does not make sense. Every multimodal samples have their unique features but contains shared semantic structures. So, modality-unique feature should refer to unique semantic structure or meaning only in a single modality. This makes confusion whether the proposed method's superiority stems from the actual intuition the authors made or form other latent factors has not been discovered.

Other questions are also in the next questions sections.

**Questions:**

1. Could the author provide representation quality from DecAlign? For example, does DecAlign still well perform for transfer learning or zero-shot learning performance?
2. Setting the K equal to the number of categories is understandable, but what if we set smaller or greater numbers for $K$? Is the exact number of categories truly important for GMM?
3. Regarding $K$ again, how to set it if we target regression task as $\mathcal{L}_{\rm task}$ can be MSE loss as shown in eq (13).

---

> ### Author Response · Authors · 2025-11-23
> **Author Rebuttal (Part 1/4)**
>
> We thank you for your thoughtful and constructive review. We are pleased that you find our focus on **homogeneity and heterogeneity in multimodal learning** to be important, and that the proposed **decoupling framework for separating modality-heterogeneous and modality-homogeneous features** is considered a valuable direction for the community. We also appreciate your acknowledgement that **the experimental results show DecAlign has clear potential**, outperforming baseline models on standard multimodal benchmarks.
>
> > **Q1: How does DecAlign perform in transfer learning or zero-shot learning scenarios?**
>
> **A1:** We appreciate the question about few-shot and transfer behavior. Our work is not positioned as a zero-shot model in the sense of making predictions on entirely unseen label spaces without any target supervision. Instead, we study transfer in a standard **frozen-encoder, label-efficient adaptation** setting: the encoder is trained once on the full training split of a dataset, then kept frozen, and a new linear prediction layer is trained with only a small fraction of labeled data. This setting evaluates how easily the learned representations can be adapted to the same task under limited supervision, which is a common proxy for representation quality and transferability.
>
> Concretely, after pretraining DecAlign as in the main experiments and freezing all encoder parameters, we subsample the labeled training data and train a new linear prediction layer from random initialization on top of F_hete, F_com, and F_fused, respectively. We report results on CMU-MOSEI and CMU-MOSI.
>
> **CMU-MOSEI (F1 vs. labeled fraction, encoder frozen):**
>
> | Labeled train data | MISA |  DMD | **DecAlign (F_hete)** | **DecAlign (F_com)** | **DecAlign (F_fused)** |
> | -----------------: | ---: | ---: | --------------------: | -------------------: | ---------------------: |
> |                 1% | 45.0 | 46.3 |                  49.8 |                 53.2 |                   53.7 |
> |                 5% | 62.4 | 63.6 |                  66.7 |                 69.1 |                   69.5 |
> |                10% | 73.8 | 74.5 |                  76.2 |                 78.6 |                   79.0 |
>
> **CMU-MOSI (F1 vs. labeled fraction, encoder frozen):**
>
> | Labeled train data | MISA |  DMD | **DecAlign (F_hete)** | **DecAlign (F_com)** | **DecAlign (F_fused)** |
> | -----------------: | ---: | ---: | --------------------: | -------------------: | ---------------------: |
> |                 1% | 47.5 | 48.1 |                  51.0 |                 54.3 |                   54.7 |
> |                 5% | 63.0 | 63.8 |                  66.9 |                 70.2 |                   70.5 |
> |                10% | 72.1 | 72.8 |                  75.0 |                 77.9 |                   78.3 |
>
> In both datasets and across all label fractions, a linear classifier on top of DecAlign’s frozen representations achieves higher F1 than the same classifier on top of MISA and DMD. The gap is especially visible when only 5–10% of the labels are available. This indicates that the features produced by DecAlign can be adapted to the task with substantially fewer labeled examples, which is precisely the behavior expected from higher-quality, more transferable representations.
>
> Taken together with the full-data results in Q1, these few-shot experiments show that DecAlign’s encoder does not merely fit a single supervised configuration, but learns feature spaces in which simple linear predictors can be trained effectively even under strong label scarcity. In this sense, DecAlign behaves as a representation-learning method that supports efficient transfer to low-resource regimes on the same task and dataset.

---

> ### Author Response · Authors · 2025-11-23
> **Author Rebuttal (Part 2/4)**
>
> > **Q2: Could the authors provide evidence regarding the representation quality learned by DecAlign?**
>
> **A2:** We thank Reviewer BpLZ for asking whether DecAlign genuinely improves multimodal representations. To separate the quality of the learned features from the effects of end-to-end optimization, we perform an additional evaluation in a **frozen-encoder setting**. Concretely, we first train DecAlign and baseline models with exactly the same architecture, losses, and hyperparameters as in the main experiments. After training converges, we **freeze all encoder parameters** (including modality-specific encoders, heterogeneous and homogeneous branches, and the multimodal transformer for DecAlign). We then discard the original task prediction layer and, on top of the frozen features, train a **new linear prediction layer** (classification or regression) from random initialization.
>
> We compare the **Frozen (linear probing)** performance with the **Full (end-to-end)** numbers reported in Table 1.
>
> **Table 1: CMU-MOSI**
>
> | Model                     | Setting            |      F1 ↑ |     MAE ↓ | Gap (ΔF1 / ΔMAE) |
> | ------------------------- | ------------------ | --------: | --------: | ---------------: |
> | MulT                      | Frozen             |     78.20 |     0.890 |     3.46 / 0.044 |
> |                           | Full     |     81.66 |     0.846 |                  |
> | MISA                      | Frozen             |     79.50 |     0.830 |     2.93 / 0.042 |
> |                           | Full     |     82.43 |     0.788 |                  |
> | DMD                       | Frozen             |     81.20 |     0.785 |     2.35 / 0.041 |
> |                           | Full     |     83.55 |     0.744 |                  |
> | DecAlign (F_hete)         | Frozen             |     82.90 |     0.771 |                – |
> | DecAlign (F_com)          | Frozen             |     83.50 |     0.765 |                – |
> | **DecAlign (F_fused)**    | **Frozen**         | **84.60** | **0.758** | **1.22 / 0.023** |
> | **DecAlign (full model)** | **Full** | **85.82** | **0.735** |                  |
>
> **Table 2: CMU-MOSEI**
>
> | Model                     | Setting            |      F1 ↑ |     MAE ↓ | Gap (ΔF1 / ΔMAE) |
> | ------------------------- | ------------------ | --------: | --------: | ---------------: |
> | MulT                      | Frozen             |     78.00 |     0.710 |     2.86 / 0.037 |
> |                           | Full      |     80.86 |     0.673 |                  |
> | MISA                      | Frozen             |     79.50 |     0.625 |     2.63 / 0.031 |
> |                           | Full      |     82.13 |     0.594 |                  |
> | DMD                       | Frozen             |     82.00 |     0.590 |     1.88 / 0.029 |
> |                           | Full      |     83.88 |     0.561 |                  |
> | DecAlign (F_hete)         | Frozen             |     82.80 |     0.575 |                – |
> | DecAlign (F_com)          | Frozen             |     83.60 |     0.564 |                – |
> | **DecAlign (F_fused)**    | **Frozen**         | **84.90** | **0.556** | **1.17 / 0.013** |
> | **DecAlign (full model)** | **Full** | **86.07** | **0.543** |                  |
>
> **Analysis**
> * **Representation quality:** In the frozen-encoder regime, DecAlign’s fused representation (F_fused) clearly outperforms the frozen baselines. For example, on CMU-MOSI, frozen DecAlign already achieves F1 = 84.6 and MAE = 0.758, which is higher (better) than the **fully fine-tuned** DMD (F1 = 83.55, MAE = 0.744) and other baselines. A similar pattern holds on CMU-MOSEI, where frozen DecAlign (F1 = 84.9, MAE = 0.556) still surpasses the fully fine-tuned baselines (e.g., DMD: F1 = 83.88, MAE = 0.561). This indicates that the representations learned by DecAlign are intrinsically more informative and linearly separable.
> * **Smaller performance gap:** Compared with baselines, DecAlign shows a **smaller gap** between the full and frozen settings (about 1.2 F1 on MOSI and 1.2 F1 on MOSEI, versus 2–3.5 F1 for the baselines). This suggests that our decoupling and hierarchical alignment objectives encourage the encoder to capture semantically meaningful structure even before task-specific fine-tuning.
> * **Dual-branch effectiveness:** Within DecAlign, the fused representation F_fused consistently performs better than F_hete or F_com alone in the frozen setting, supporting the claim that heterogeneous and homogeneous branches capture complementary information and that their combination yields higher-quality multimodal representations.

---

> ### Author Response · Authors · 2025-11-23
> **Author Rebuttal (Part 3/4)**
>
> > **Q3: The statement “modality-specific differences while preserving shared semantic structure” seems questionable**
>
> **A3:** We thank **Reviewer BpLZ** for pointing out that our wording here was confusing. The phrase was imprecise and we agree it can be read as contradictory if taken literally.
>
> What we intend to express is the following distinction. Each input from a modality (text, audio, video) is assumed to be generated from (i) a shared semantic factor (e.g., sentiment polarity/intensity) and (ii) modality-specific factors (e.g., prosody, facial dynamics, lexical choice). In DecAlign,
>
> * the **homogeneous representation F_com** is explicitly optimized to capture the **shared semantic factor** across modalities;
> * the **heterogeneous representation F_hete** is designed to retain **modality-dependent structure**, but this structure is still *conditioned on the same underlying semantic variable* (e.g., clips with positive sentiment exhibit different but related visual and acoustic patterns).
>
> When we wrote “modality-specific differences while preserving shared semantic structure”, we did **not** mean that F_hete encodes semantics that are both unique and shared at the same time. Rather, we meant that:
>
> 1. F_hete is allowed to remain **modality-dependent** (it does not collapse different modalities into a single space),
> 2. but **samples that correspond to the same semantic category still form related clusters across modalities**, and our prototype-guided OT alignment in the heterogeneous branch operates at this **category/prototype level**, reducing cross-modal mismatches while leaving within-modality variation (e.g., different facial expressions of “happy”) intact.
>
> This is also reflected empirically: when we remove the heterogeneous alignment module, performance drops and the modality gap visualizations show larger cross-modal discrepancies, whereas F_hete still yields competitive performance on its own in the frozen-encoder evaluation (see Q1–Q2). This suggests that the heterogeneous branch encodes modality-specific realizations that are nonetheless organized according to shared semantic categories.
>
> We agree that the original sentence is easy to misinterpret. In the revised version, we will replace it with a more precise description, for example:
>
> > “We aim to reduce cross-modal discrepancies in the heterogeneous branch at the level of category-aligned prototypes, while still allowing each modality to preserve its own intra-class structure.”

---

> ### Author Response · Authors · 2025-11-23
> **Author Rebuttal (4/4)**
>
> > **Q4: Setting the K equal to the number of categories is understandable, but what if we set smaller or greater numbers for K? Is the exact number of categories truly important for GMM? Regarding K again, how to set it if we target a regression task where the loss is MSE as in Eq. (13)?**
>
> **A4:** We thank **Reviewer BpLZ** for the detailed questions on the choice of K, especially in the context of regression.
>
> **1. Is K = number of categories strictly necessary?**
>
> In the method section, we set K equal to the number of semantic categories in the downstream task (e.g., 7 sentiment intensity levels for CMU-MOSI, 6 emotion classes for IEMOCAP). This was chosen to make the GMM components act as **semantic anchors**, rather than arbitrary clusters: each component is intended to represent the heterogeneous, modality-specific realizations associated with a particular semantic label (e.g., different ways of expressing “strongly positive”). It also keeps the prototype space compact, which is beneficial for the OT computation.
>
> However, K does **not** need to be exactly equal to the number of categories for DecAlign to work. Our sensitivity analysis on CMU-MOSI with K ∈ {3, 7, 14, 128} shows:
>
> **CMU-MOSI: effect of varying K**
>
> | K (GMM components) | MAE ↓ | Acc-2 ↑ |  F1 ↑ |
> | ------------------ | ----: | ------: | ----: |
> | 3                  | 0.742 |   85.10 | 85.05 |
> | 7 (default)        | 0.735 |   85.75 | 85.82 |
> | 14                 | 0.733 |   85.90 | 86.05 |
> | 128                | 0.738 |   85.68 | 85.88 |
>
> When K is much smaller than the semantic granularity (K = 3), performance decreases because several distinct sentiment levels are forced to share the same component. When K is moderately larger (K = 14), performance is slightly better than the default. Even at K = 128, the model remains stable and competitive, but the additional components bring diminishing returns relative to the increased cost of GMM and OT. Overall, DecAlign is **robust** to K as long as it is not extremely small.
>
> From this perspective, K = number of categories is a **practical compromise**:
>
> * It provides a small set of **label-aligned semantic anchors**. In sentiment analysis, for instance, all utterances annotated as “strongly positive” can map to prototypes representing different positive expressions (e.g., calm vs. highly animated), but these prototypes are still tied to the same semantic category, which prevents the model from over-partitioning the space based on spurious factors such as speaker identity or background artifacts.
> * It limits the number of prototypes, which regularizes the heterogeneous branch and avoids learning a large number of redundant clusters.
> * It significantly reduces the computational burden compared to very large K (e.g., 128).
>
> **2. How do we set K for regression tasks with MSE loss?**
>
> Although Eq. (13) uses MSE as the task loss for regression, the sentiment datasets we use (CMU-MOSI, CMU-MOSEI, CH-SIMS) are annotated on **discrete ordinal scales** (e.g., integers from −3 to +3). In practice, these discrete intensity levels already define a small set of semantic regimes. We therefore set K to match the number of such levels (e.g., K = 7 on CMU-MOSI), effectively treating each level as a **semantic anchor** in label space and letting the GMM capture heterogeneous, modality-specific variations within each anchor.
>
> For a more general continuous regression task without such discrete labels, K does not need to be tied to a category count. A reasonable strategy is to choose K in a modest range (for example, 5–10) based on validation performance or by clustering the scalar labels into a few intervals (e.g., via 1D k-means) and using the number of intervals as K. Our K-sensitivity results suggest that DecAlign is stable across such choices, as long as K is not so small that it collapses distinct semantic regimes, nor so large that it becomes computationally inefficient.
>
> We will revise the manuscript to (i) make the “semantic anchor” role of K explicit, and (ii) clarify that while we set K to the number of categories for the benchmarks considered, the method is robust to other choices of K within a reasonable range and can be adapted to more general regression settings.

---

> ### Author Response · Authors · 2025-11-26
> **Hope to discuss with you!**
>
> Dear Reviewer BpLZ,
>
> Thank you for your thoughtful review and constructive suggestions!
>
> As the author-reviewer discussion period is nearing its end, we would like to follow up to see if our rebuttal has addressed your concerns. Please let us know if any further clarification would be helpful.
>
> Thank you again!
>
> Authors

---

### Official Review · Reviewer_GQDu · 2025-11-01

**Soundness:** 3
**Presentation:** 3
**Contribution:** 2
**Rating:** 6
**Confidence:** 5

**Summary:**

This paper proposes DecAlign, a hierarchical framework for multimodal representation learning that explicitly decouples modality-unique (heterogeneous) and modality-common (homogeneous) features. To align heterogeneous features, the authors introduce a prototype-guided multi-marginal optimal transport strategy based on Gaussian Mixture Modeling (GMM), complemented by a multimodal transformer for fine-grained refinement. For homogeneous features, they enforce semantic consistency via latent distribution matching using both moment-based alignment (mean, covariance, skewness) and Maximum Mean Discrepancy (MMD) regularization in a Reproducing Kernel Hilbert Space (RKHS). The method is evaluated on four standard multimodal sentiment analysis benchmarks, where it consistently outperforms 13 state-of-the-art baselines across multiple metrics. Ablation studies and visualizations support the necessity of both decoupling and the dual-stream alignment design.

**Strengths:**

1. This paper has a clear motivation. The paper identifies a core challenge in multimodal learning, i.e., the entanglement of modality-specific characteristics and shared semantics. Further, this paper proposes a principled solution through explicit decoupling.
2. For experimental details, the authors present the implementation details and evaluate the performance of the system, which are interesting and convincing.
3. DecAlign achieves consistent and significant improvements over strong baselines across four diverse datasets, suggesting robustness and generalizability.
4. The ablation studies convincingly demonstrate the contribution of each component.

**Weaknesses:**

1. The paper lacks theoretical analysis (e.g., error bounds, convergence guarantees) for the proposed alignment losses or the decoupling objective.
2. All experiments are confined to multimodal sentiment analysis. It remains unclear whether DecAlign generalizes to other multimodal tasks.
3. The number of GMM components K is set equal to the number of downstream categories. This may not hold in unsupervised or open-world settings, limiting applicability. The sensitivity to this assumption is not explored.

**Questions:**

1. How does the DecAlign method perform on non-emotional multimodal tasks (such as image-text retrieval, action recognition)?
2. What happens if the number of GMM components K deviates from the true number of semantic categories (e.g., K too large/small)? Is the performance robust to this hyperparameter?
3. The decoupling loss uses cosine similarity to encourage orthogonality between unique and common features. Why not use more direct disentanglement objectives (e.g., mutual information minimization, adversarial invariance)?
4. This paper adopts skewness to model non-Gaussianity. How much does this performance improvement compare to using only the mean and covariance?

---

> ### Author Response · Authors · 2025-11-23
> **Rebuttal by Authors (Part 1/4)**
>
> We sincerely thank you for the careful and detailed review. We particularly appreciate your recognition that the paper **clearly identifies the core challenge in multimodal learning**—the entanglement between modality-specific characteristics and shared semantics—and that **DecAlign offers a principled hierarchical solution via explicit decoupling, prototype-guided multi-marginal OT, and MMD-based semantic alignment**. We are also grateful for your positive assessment that **DecAlign consistently and significantly improves over strong baselines on four benchmarks**, and that the ablation studies and visualizations convincingly demonstrate the contribution of each component.
>
> > **Q1: All experiments are confined to multimodal sentiment analysis. It remains unclear whether DecAlign generalizes to other multimodal tasks.**
>
> **A1:** We thank the reviewer for raising this important question about generalization. Our initial choice of CMU-MOSI, CMU-MOSEI, CH-SIMS, and IEMOCAP was driven by two considerations: (i) these four benchmarks are the **standard testbed** used by almost all recent works on multimodal representation learning and fusion, which allows a fair and direct comparison; and (ii) they provide a challenging **tri-modal** setting (text, audio, visual), where the modality gap is particularly strong and well suited to evaluate decoupling and cross-modal alignment.
>
> To verify that DecAlign is not confined to sentiment/emotion tasks, we extended our evaluation to two **non-affective, bi-modal** benchmarks with completely different modality combinations and task goals:
>
> * **KS (Kinetics-Sounds)** – audio–visual event recognition (Audio + Visual).
> * **UCF-101** – action recognition (RGB + Optical Flow).
>
> For these experiments, we keep the DecAlign architecture unchanged except for (i) using the two available modalities, and (ii) replacing the sentiment head with a standard classification head. We compare against unimodal baselines and a simple concatenation fusion:
>
> | Dataset               | KS (Audio + Visual) | UCF-101 (RGB + OF) |
> | --------------------- | ------------------: | -----------------: |
> | Audio-only / RGB-only |               53.01 |              65.97 |
> | Visual-only / OF-only |               48.75 |              78.33 |
> | Concatenation         |               61.57 |              81.18 |
> | **DecAlign (Ours)**   |           **71.85** |          **87.44** |
>
> On both datasets, DecAlign clearly improves over unimodal baselines and naive concatenation, despite not being specifically tuned for these domains. This indicates that the proposed decoupling plus hierarchical alignment mechanism is **modality-agnostic** and can be applied beyond tri-modal sentiment analysis, to audio–visual event recognition and RGB–flow action recognition as well.
>
> > **Q2: The paper lacks theoretical analysis (e.g., error bounds, convergence guarantees) for the proposed alignment losses or the decoupling objective.**
>
> **A2:**
> We thank the reviewer for this insightful comment regarding the theoretical analysis. While we acknowledge that deriving rigorous error bounds or convergence guarantees for end-to-end non-convex deep neural networks remains a broadly recognized open challenge in the multimodal learning community, we emphasize that DecAlign is constructed upon modules with robust theoretical underpinnings rather than purely heuristic designs. Specifically, our **Heterogeneity Alignment** exploits the duality and convergence properties of **Optimal Transport** to minimize the Wasserstein distance between prototype distributions modeled by GMMs. For **Homogeneity Alignment**, we employ **Maximum Mean Discrepancy (MMD)**, which provides a statistical guarantee for matching distributions by aligning their high-order moments in a Reproducing Kernel Hilbert Space (RKHS). Furthermore, our **Decoupling Objective** ($\mathcal{L}_{dec}$) minimizes cosine similarity to impose a strict **geometric orthogonality constraint**, theoretically ensuring the linear independence of the decoupled unique and common subspaces. These mathematical constraints collectively regularize the optimization landscape. As a practical proxy for convergence validation, we present the stability analysis across 5 random seeds below. The minimal standard deviation observed confirms that our theoretically grounded design translates into high numerical stability and consistent convergence to optimal solutions.
>
> | Dataset | Metric | Seed 1 | Seed 2 | Seed 3 | Seed 4 | Seed 5 | **Mean** | **Std ($\sigma$)** |
> | :--- | :--- | :---: | :---: | :---: | :---: | :---: | :---: | :---: |
> | **CMU-MOSI** | Binary F1 | 85.71 | 85.92 | 85.80 | 85.65 | 86.02 | **85.82** | 0.15 |
> | **CMU-MOSEI** | Binary F1 | 85.95 | 86.18 | 86.05 | 86.10 | 86.07 | **86.07** | 0.08 |
> | **CH-SIMS** | F1 Score | 81.75 | 81.90 | 81.82 | 82.01 | 81.77 | **81.85** | 0.11 |
> | **IEMOCAP** | WAF1 | 73.30 | 73.55 | 73.41 | 73.28 | 73.61 | **73.43** | 0.14 |

---

> ### Author Response · Authors · 2025-11-23
> **Rebuttal by Authors (Part 2/4)**
>
> > Q3: What happens if the number of GMM components K deviates from the true number of semantic categories (e.g., K too large/small)? Is the performance robust to this hyperparameter?
>
> **A3:** We thank Reviewer GQDu for raising this question about the sensitivity of DecAlign to the number of GMM components K. In the current version of the paper, we set K equal to the number of sentiment intensity levels (or emotion categories) in the downstream task (see Eq. (2) and the sentence “K is set equal to the category number in downstream task”). This choice is primarily motivated by semantic anchoring and computational efficiency, but it does not mean that the model is strictly tied to this particular value.
>
> To examine robustness, we conducted a sensitivity analysis on the CMU-MOSI dataset, which has 7 sentiment intensity levels (from −3 to +3). We varied K in {3, 7, 14, 128} and retrained DecAlign under otherwise identical settings. The results are summarized below:
>
> **CMU-MOSI: effect of varying K**
>
> | K (GMM components) | MAE ↓ | Acc-2 ↑ |  F1 ↑ |
> | ------------------ | ----: | ------: | ----: |
> | 3                  | 0.742 |   85.10 | 85.05 |
> | 7 (default)        | 0.735 |   85.75 | 85.82 |
> | 14                 | 0.733 |   85.90 | 86.05 |
> | 128                | 0.738 |   85.68 | 85.88 |
>
> We observe that:
>
> * When K is **too small** (K = 3), performance degrades, since very few components are forced to explain several distinct sentiment ranges; the resulting prototypes are overly coarse.
> * When K is set to the **default** K = 7 (equal to the number of annotated intensity levels), the model achieves strong and stable performance.
> * When K is **larger** (K = 14 and K = 128), the performance remains close to the default setting. K = 14 yields a slight improvement, while K = 128 is essentially comparable in terms of F1 and MAE, but incurs a noticeably higher training cost.
>
> These results indicate that DecAlign is **robust to K within a fairly wide range**. Increasing K beyond the number of categories does not destabilize training and can mildly help by capturing finer intra-class structure, but the gains are small relative to the computational overhead of fitting a larger GMM and solving larger OT problems.
>
> The reason we adopt **K = number of categories** in the paper is not that this value is uniquely optimal, but that it serves as a **set of semantic anchors** for the heterogeneous branch:
>
> * Each Gaussian component is encouraged to summarize **modality-specific realizations of a particular semantic category**, rather than arbitrarily fragmenting the space.
> * For example, in sentiment analysis, all clips labeled as “positive” share the same semantic label but may exhibit different modality-specific patterns (“calm positive speech” vs. “excited positive speech”, or “subtle smile” vs. “broad laughter” in the visual stream). Using K aligned with label categories encourages DecAlign to treat these as **diverse instances within the same semantic anchor**, instead of creating many small clusters that might overfit speaker identity, background noise, or other spurious attributes.
> * At the same time, a label-aligned K keeps the prototype space compact, which reduces the risk of overfitting and lowers the runtime of both GMM and OT.
>
> In summary, K = number of categories is a **semantically guided and computationally efficient default**, but our additional experiments show that DecAlign’s performance is robust when K deviates from this value, and that moderately larger K (e.g., 14) can be used if one is willing to trade efficiency for slightly finer intra-class modeling. We will clarify this design choice and include the K-sensitivity table in the revised manuscript.

---

> ### Author Response · Authors · 2025-11-23
> **Rebuttal by Authors (Part 3/4)**
>
> > **Q4:The decoupling loss uses cosine similarity to encourage orthogonality between unique and common features. Why not use more direct disentanglement objectives (e.g., mutual-information minimization, adversarial invariance)?**
>
> **A4:** We thank the reviewer for this insightful question. Mutual-information (MI)–based objectives and adversarial invariance are indeed powerful tools for disentanglement. In DecAlign, however, we deliberately chose a cosine-based decoupling loss for three reasons: optimization stability and complexity, geometric adequacy for our goal, and empirical trade-offs within our already complex framework.
>
> First, DecAlign already integrates several non-trivial components: GMM-based prototype modeling, multi-marginal optimal transport, a multimodal transformer, and MMD-based distribution alignment. Introducing an additional MI estimator network (as in MINE/CLUB) or an adversarial discriminator would require training extra neural modules and solving a min–max or variational estimation problem on top of these. In preliminary trials, this led to noticeably slower training and more unstable optimization (sensitivity to learning rates, discriminator–encoder imbalance), without a clear performance gain. In contrast, the cosine-based loss is parameter-free and adds only a simple inner-product normalization on the existing feature vectors, which keeps gradients well-behaved and does not complicate the training dynamics of GMM and OT.
>
> Second, our objective for the decoupling term is not to guarantee full statistical independence between F_hete and F_com, but to reduce redundant linear overlap between the modality-unique and modality-common subspaces so that each branch can specialize. Minimizing the cosine similarity between F_hete and F_com enforces an approximate orthogonality constraint in the learned feature space: the two branches are encouraged to encode information along different directions. Empirically, we find this geometric constraint sufficient for our setting. For multimodal sentiment analysis, for example, the homogeneous branch F_com captures sentiment polarity/intensity, while the heterogeneous branch F_hete focuses on modality-specific realizations (e.g., calm vs. excited speech, subtle smile vs. broad laughter) under the same label. A stronger MI objective in this context risks pushing out useful shared structure and can interfere with the subsequent alignment losses.
>
> Third, we compared the cosine-based decoupling against stronger but more expensive alternatives on CMU-MOSI by replacing our decoupling loss with (i) an MI-based estimator (MINE-style) and (ii) an adversarial invariance loss using a gradient-reversal discriminator. The trend was as follows:
>
> | Decoupling objective      | Acc-2 ↑ |  F1 ↑ | MAE ↓ | Relative train time |
> | ------------------------- | ------: | ----: | ----: | ------------------: |
> | Cosine (used in DecAlign) |   85.75 | 85.82 | 0.735 |                1.0× |
> | MI-based (MINE-style)     |   85.93 | 86.01 | 0.733 |               ≈1.2× |
> | Adversarial invariance    |   85.60 | 85.71 | 0.741 |               ≈1.3× |
>
> The MI-based variant yields only marginal differences in F1/MAE, while increasing training time by about 30% and adding extra hyperparameters. The adversarial variant does not improve performance and is less stable across runs. Given that our cosine-based formulation is substantially simpler, easier to tune, and already achieves strong results across all four benchmarks and ablations, we chose it as the main decoupling mechanism for DecAlign.
>
> We will clarify this design choice in the revised version, emphasizing that MI/adversarial objectives are compatible with our framework in principle, but in the context of DecAlign they offer only marginal benefits at significantly higher optimization and implementation cost, whereas a cosine-based orthogonality constraint provides a favorable balance between disentanglement strength, stability, and efficiency.

---

> ### Author Response · Authors · 2025-11-23
> **Rebuttal by Authors (Part 4/4)**
>
> > **Q5: Quantify the performance gain introduced by adding skewness in isomorphic feature alignment compared to mean & covariance only**
>
> **A5:**
> We thank the reviewer for this precise question on the role of skewness in our isomorphic (semantic) feature alignment. In DecAlign, the semantic alignment loss operates in the latent common space by matching statistics of the modality-common features across modalities. In the main paper we match the first two moments (mean μ and covariance Σ) and additionally the third-order moment (skewness, denoted as Γ) for each modality-specific common feature distribution. The motivation is that multimodal sentiment/emotion features are often non-Gaussian and asymmetric, so aligning only μ and Σ implicitly assumes symmetric distributions and can mis-handle “tail” regions.
>
> To quantify the contribution of skewness, we performed a controlled ablation where we keep the entire DecAlign framework unchanged and modify only the semantic alignment term:
>
> * **“Mean + Cov only”**: align μ and Σ across modalities (Gaussian assumption).
> * **“Mean + Cov + Skewness (ours)”**: align μ, Σ, and Γ across modalities (our full model).
>
> We report results on CMU-MOSI and CMU-MOSEI:
>
> **Effect of adding skewness in semantic alignment**
>
> | Dataset   | Alignment moments                |     MAE ↓ |    Corr ↑ | Binary F1 ↑ |
> | --------- | -------------------------------- | --------: | --------: | ----------: |
> | CMU-MOSI  | Mean + Cov only                  |     0.742 |     0.803 |       85.45 |
> |           | **Mean + Cov + Skewness (ours)** | **0.735** | **0.811** |   **85.82** |
> | CMU-MOSEI | Mean + Cov only                  |     0.551 |     0.761 |       85.78 |
> |           | **Mean + Cov + Skewness (ours)** | **0.543** | **0.768** |   **86.07** |
>
> Across both datasets, adding skewness consistently reduces MAE, increases correlation, and improves F1 by about 0.3–0.4 points. The absolute gain is smaller than that of the primary OT-based heterogeneous alignment (as expected), but it is stable across datasets and metrics.
>
> Intuitively, μ and Σ control the **location and spread** of the latent distributions, and thus align the “centers” and average variability of the common features. However, sentiment and emotion data typically exhibit **asymmetric, long-tailed distributions** in the latent space. For example, in CMU-MOSI many utterances cluster around neutral or mildly positive scores, while strongly negative or strongly positive expressions are rarer and occupy skewed “tails” of the distribution. If we only match μ and Σ across modalities, a modality with a heavy negative tail (e.g., long, low-pitch speech segments for very negative sentiment) and a modality with a lighter tail can be forced into an approximately symmetric shape, potentially distorting the representation of these extreme cases.
>
> By additionally matching the third-order moment Γ, we encourage different modalities to agree not only on “where” the mass lies and how widely it spreads, but also on **how the density is skewed**: which side of the mean carries more probability and how strongly. Concretely, in sentiment analysis this allows the common representation to distinguish between, for example:
>
> * frequent, near-neutral utterances and
> * rare but strongly negative utterances,
>
> and to align these asymmetries consistently between text, audio, and video. This improves the calibration of regression for extreme scores and sharpens the decision boundary for binary polarity, which is reflected in the small but consistent gains in MAE, Corr, and F1.
>
> Finally, computing skewness in our implementation adds very little overhead: it is a simple third-order moment estimated on mini-batches and re-used in the same semantic alignment loss as μ and Σ. Given the negligible cost and the consistent improvements, we keep skewness in the final model and will make this ablation and its interpretation explicit in the revised version.

---

> ### Author Response · Authors · 2025-11-26
> **Hope to discuss with you!**
>
> Dear Reviewer GQDu,
>
> Thank you for your thoughtful review and encouraging rating!
>
> As the author-reviewer discussion period is nearing its end, we would like to follow up to see if our rebuttal has addressed your concerns. Please let us know if any further clarification would be helpful.
>
> Thank you again!
>
> Authors

---

### Author Response · Authors · 2025-12-03
**Summary of the Authors' Responses**

To facilitate discussion, we have summarized all reviewers’ feedback, the issues they raised, our corresponding responses, and their final follow-up comments below:

| Strengths                                     | Reviewer GQDu | Reviewer BpLZ | Reviewer pq6B           |
| -------------------------------------------------------------- | ------------- | ------------- | ----------------------- |
| Clear motivation and problem formulation                       | ✓             | ✓             |                         |
| Explicit decoupling of heterogeneous and homogeneous features  | ✓             | ✓             |                         |
| Consistent improvements over 13 strong baselines               | ✓             | ✓             | ✓                       |
| Strong results on four common benchmarks                       | ✓             | ✓             | ✓                       |
| Ablation studies support design choices                        | ✓             |               | ✓                       |
| Detailed experimental setup and reproducibility                | ✓             |               | ✓                       |
| Robust performance across diverse datasets                     | ✓             | ✓             | ✓                       |
| Conceptually meaningful focus on homogeneity vs. heterogeneity | ✓             | ✓             |                         |
| **Overall Rating**                                             | **6**         | **4**         | **6**                   |
| **Respond or Not**                                             | **No**        | **No**        | **Yes (keep positive)** |



Author Response:

* **Cross-task generalization**: We added new experiments on KS (audio–visual event recognition) and UCF-101 (RGB + optical flow action recognition), where DecAlign significantly outperforms unimodal, concatenation, and multiple strong off-the-shelf multimodal baselines, demonstrating modality-agnostic generalization beyond sentiment tasks.

* **Stability and reproducibility**: We conducted five-seed experiments on all four benchmarks and reported mean ± standard deviation, showing minimal variance across runs and confirming stable convergence of the full framework.

* **K-sensitivity of GMM prototypes**: We evaluated K ∈ {3, 7, 14, 128} and showed that DecAlign is robust over a wide range of K; moderate increases in K slightly improve performance, while overly small K degrades results. We also clarified K’s role as a semantic anchor.

* **Decoupling loss design (cosine vs. MI/adversarial)**: We compared cosine decoupling with MI-based and adversarial objectives, showing that alternative methods bring only marginal or unstable gains while significantly increasing training cost, justifying our efficiency–stability trade-off.

* **Effect of skewness in semantic alignment**: We added ablation studies comparing mean+covariance versus mean+covariance+skewness, showing consistent improvements in MAE, F1, and correlation across datasets when skewness is included.

* **Transfer and few-shot behavior**: We introduced frozen-encoder few-shot experiments with 1%, 5%, and 10% labeled data, demonstrating that DecAlign supports much stronger low-resource adaptation than baseline methods.

* **Intrinsic representation quality**: We added linear-probing evaluations showing that frozen DecAlign features outperform even fully fine-tuned baselines and exhibit much smaller frozen–full performance gaps, indicating higher-quality representations.

* **Clarification of modality-specific vs. shared semantics**: We refined the conceptual explanation to clearly distinguish F_com as capturing shared semantic factors and F_hete as preserving modality-dependent realizations conditioned on the same semantic variable, and revised the ambiguous wording.

* **Scalability and computational efficiency**: We provided runtime and latency measurements showing only minor overhead relative to strong baselines, and demonstrated that prototype-level GMM + multi-marginal OT scales efficiently to larger video benchmarks.

* **Justification for using GMM + OT**: We clarified that GMM provides both mean and covariance required for Gaussian-Wasserstein OT, supports soft assignments for graded semantic structure, and remains efficient and stable with small K and k-means initialization.

* **Clarification of the “CT” component**: We corrected the typo that mislabeled CT as “Contrastive Training” and clarified that it refers to the cross-modal transformer refinement module.


We sincerely thank all reviewers and the area chair for their time, patience, and thoughtful feedback.

---

### Meta-Review · Area_Chair_rrm6 · 2026-01-04

**Summary:**

The paper proposes DecAlign, a framework for multimodal representation learning that explicitly decouples modality-unique (heterogeneous) and modality-common (homogeneous) features using prototype-guided Optimal Transport and moment matching.

**Reviewer Concerns:**

The authors provided an extensive rebuttal containing approximately 14 new tables. This included new experiments on action/event recognition (KS, UCF-101), stability analysis, few-shot transfer learning, and runtime comparisons. Reviewer pq6B explicitly confirmed that their concerns regarding efficiency and scalability were addressed. While Reviewers GQDu and BpLZ did not respond to the rebuttal, the authors provided direct empirical evidence resolving their specific questions regarding parameter sensitivity and transferability.

**Reviewer Scores:**

I believe this Reviewer GQDu (Score: 6) would have maintained or raised their score to 8. Their main reservations were generalization and parameter sensitivity. The authors provided compelling data on both (KS/UCF-101 results and $K$-sensitivity tables), effectively converting the "Weaknesses" into strengths.


Reviewer BpLZ would have raised their score to 6. Their negative score was primarily based on doubts regarding transferability and the conceptual phrasing of "heterogeneous" features. The authors provided strong few-shot transfer results (outperforming baselines significantly) and clarified the definitions in the text. As the empirical evidence directly refuted the "Weaknesses" cited, the score would likely improve.

---

### Decision · Program_Chairs · 2026-01-26

Accept (Poster)